# No Free Lunch from Random Feature Ensembles

## Abstract

Given a budget on total model size, one must decide whether to train a single, large neural network or to combine the predictions of many smaller networks. We study this trade-off for ensembles of random-feature ridge regression models. We prove that when a fixed number of trainable parameters are partitioned among $K$ independently trained models, $K = 1$ achieves optimal performance, provided the ridge parameter is optimally tuned. We then derive scaling laws which describe how the test risk of an ensemble of regression models decays with its total size. We identify conditions on the kernel and task eigenstructure under which ensembles can achieve near-optimal scaling laws. Training ensembles of deep convolutional neural networks on CIFAR-10 and a transformer architecture on C4, we find that a single large network outperforms any ensemble of networks with the same total number of parameters, provided the weight decay and feature-learning strength are tuned to their optimal values.

## 1 Introduction

Ensembling methods are a well-established tool in machine learning for reducing the variance of learned predictors. While traditional ensemble approaches like random forests (Breiman, 2001) and XGBoost (Chen & Guestrin, 2016) combine many weak predictors, the advent of deep neural networks has shifted the state of the art toward training a single large predictor (LeCun et al., 2015). However, deep neural networks still suffer from various sources of variance, such as finite datasets and random initialization (Atanasov et al., 2022; Adlam & Pennington, 2020; Lin & Dobriban, 2021; Atanasov et al., 2024). As a result, deep ensembles—ensembles of deep neural networks—remain a popular method for variance reduction (Ganaie et al., 2022; Fort et al., 2020), and uncertainty estimation (Lakshminarayanan et al., 2017).

A critical consideration in practice is the computational cost associated with ensemble methods. While increasing the number of predictors in an ensemble improves its accuracy (provided each ensemble member is "competent" (Theisen et al., 2023)), each additional model incurs significant computational overhead. Supposing a fixed memory capacity for learned parameters, a more pragmatic comparison is between an ensemble of neural networks and a single large network with the same total parameter count. Indeed, recent studies have called into question the utility of an ensemble of deep networks relative to a single network of comparable total parameter count (Abe et al., 2022; Vyas et al., 2023).

Originally introduced as a fast approximation to Kernel Ridge Regression, random-feature ridge regression (RFRR) Rahimi & Recht (2007) has emerged as a rich "toy model" for deep learning, capturing non-trivial effects of dataset size and network width (Canatar et al., 2021; Atanasov et al., 2023; Mei & Montanari, 2022). While over-fitting effects known as "double-descent" may lead to non-monotonic behavior of the loss in both deep networks and ridge regression (D'Ascoli et al., 2020; Nakkiran, 2019; Nakkiran et al., 2019; Adlam & Pennington, 2020; Lin & Dobriban, 2021), double-descent can be mitigated by optimally tuning the ridge parameter (Nakkiran et al., 2020; Advani et al., 2020; Canatar et al., 2021; Simon et al., 2023). Specifically, Simon et al. showed that in RFRR, test risk decreases monotonically with model size and dataset size when the ridge parameter is optimally chosen.

In this present work, we study the tradeoff between the number of predictors and the size of each predictor in ensembles of RFRR models. We find that with a fixed total parameter budget, minimal

error is achieved by a single predictor trained on the full set of features, provided the ridge parameter is optimally tuned. We analyze the trade-off between ensemble size and model size for tasks with a power-law eigenstructure, identifying regimes where near-optimal performance can still be achieved with an ensemble. Finally, we present experiments suggesting that this "no free lunch from ensembles" principle extends to deep feature-learning ensembles when network hyperparameters are finely tuned. The remainder of this work is organized as follows:

In section 2 we review the necessary background on the theory and practice of RFRR and its extension to ensembles.

In section 3, we state an omniscient risk estimate for ensembled RFRR and extend the "bigger is better" theorem of Simon et al. (2023) to the ensembled case.

In Section 4, we prove that, when the total number of features is fixed, the optimal test risk of an ensemble of RFRR models is achieved when the available features are consolidated into a single large model, provided that the ridge parameter is tuned to its optimal value. We confirm these predictions for ReLU RFRR models on binarized CIFAR-10 and MNIST classification tasks.

In Section 5, we derive scaling laws for the test risk of an ensemble of RFRR models under source and capacity constraints (Cui et al., 2023; Caponnetto & De Vito, 2006; Bordelon et al., 2020; Defilippis et al., 2024), in the width-bottlenecked regime. We identify regimes where *near*-optimal performance can be achieved using an ensemble of smaller predictors. The derived scaling laws provide a good description of RFRR on the binarized CIFAR-10 and MNIST classification tasks.

In Section 6, we test whether the intuitions provided by random feature models carry over to deep neural networks in computer vision and natural language processing tasks. We find that for deep networks in both the lazy and rich feature-learning regimes, a single large network typically outperforms an ensemble of smaller networks, provided that the weight decay is tuned to its optimal value.

## 2 PRELIMINARIES

### 2.1 RANDOM FEATURES AND THE KERNEL EIGENSPECTRUM

In this section, we describe ensembled RFRR, as well as the spectral decomposition of the kernel on which our results rely. This framework is described in (Simon et al., 2023) for the single-predictor case, and reviewed in more rigor in Appendix A.

**Kernel Ridge Regression.** In standard kernel ridge regression, the goal is to learn a function $f(\boldsymbol{x})$ that maps input features $\boldsymbol{x} \in \mathbb{R}^D$ to a target value $y \in \mathbb{R}$, given a training set $\mathcal{D} = \{\boldsymbol{x}_p, y_p\}_{p=1}^P$. The learned function can be expressed in terms of the kernel function $\boldsymbol{H}(\boldsymbol{x}, \boldsymbol{x}') : \mathbb{R}^D \times \mathbb{R}^D \mapsto \mathbb{R}$ as:

$$f(\boldsymbol{x}) = \boldsymbol{h}_{x,\mathcal{X}} \left(\boldsymbol{H}_{\mathcal{X}\mathcal{X}} + \lambda \boldsymbol{I}\right)^{-1} \boldsymbol{y}, \tag{1}$$

where $\boldsymbol{H}_{\mathcal{X}\mathcal{X}} \in \mathbb{R}^{P \times P}$ is the kernel matrix with entries $[\boldsymbol{H}_{\mathcal{X}\mathcal{X}}]_{pp'} = \boldsymbol{H}(\boldsymbol{x}_p, \boldsymbol{x}_{p'})$, and $\boldsymbol{h}_{x,\mathcal{X}} = [\boldsymbol{H}(\boldsymbol{x}, \boldsymbol{x}_1), \ldots, \boldsymbol{H}(\boldsymbol{x}, \boldsymbol{x}_P)]$. The vector $\boldsymbol{y} \in \mathbb{R}^P$ contains the training labels, and $\lambda$ is the ridge parameter.

This procedure can be viewed as performing linear regression in the RKHS defined by the kernel. Specifically, the kernel $\boldsymbol{H}(\boldsymbol{x}, \boldsymbol{x}')$ can be decomposed into its eigenfunctions $\{\phi_t(\boldsymbol{x})\}_{t=1}^{\infty}$ and corresponding eigenvalues $\{\eta_t\}_{t=1}^{\infty}$:

$$\boldsymbol{H}(\boldsymbol{x}, \boldsymbol{x}') = \sum_{t=1}^{\infty} \eta_t \phi_t(\boldsymbol{x}) \phi_t(\boldsymbol{x}'). \tag{2}$$

In this formulation, $f(\boldsymbol{x})$ is equivalent to the function learned by linear regression in the infinite-dimensional feature space with dimensions given by $\theta_t(\boldsymbol{x}) \equiv \sqrt{\eta_t}\phi_t(\boldsymbol{x})$. Similarly, we will assume that the target function can be decomposed in this basis as $f_*(\boldsymbol{x}) = \sum_t \bar{\boldsymbol{w}}_t \theta_t(\boldsymbol{x})$. The training labels are assigned as $y_p = f_*(\boldsymbol{x}_p) + \epsilon_p$ where $\epsilon_p \sim \mathcal{N}(0, \sigma_\epsilon^2)$ is drawn i.i.d. for each sample.

**Random-Feature Ridge Regression (RFRR).**    An approximation of kernel ridge regression may be achieved by mapping the input data into a finite-dimensional feature space, where linear regression is performed. Consider the "featurization" transformation $g : \mathbb{R}^C \times R^D \mapsto \mathbb{R}$. Define the random features $\boldsymbol{\psi}(\boldsymbol{x}) \in \mathbb{R}^N$ by $[\boldsymbol{\psi}(\boldsymbol{x})]_n = g(\boldsymbol{v}_n, \boldsymbol{x})$ for independently drawn $\boldsymbol{v}_n \sim \mu_{\boldsymbol{v}}$. The prediction of the RFRR model is

$$f(\boldsymbol{x}) = \boldsymbol{w}^\top \boldsymbol{\psi}(\boldsymbol{x}), \tag{3}$$

where $\boldsymbol{w}$ is the weight vector learned via ridge regression. The random features model can be interpreted as kernel ridge regression with a stochastic kernel $\hat{\boldsymbol{H}}(\boldsymbol{x}, \boldsymbol{x}')$, defined as:

$$\hat{\boldsymbol{H}}(\boldsymbol{x}, \boldsymbol{x}') = \frac{1}{N} \sum_{n=1}^N g(\boldsymbol{v}_n, \boldsymbol{x}) g(\boldsymbol{v}_n, \boldsymbol{x}'). \tag{4}$$

As $N \to \infty$, this stochastic kernel converges to the deterministic kernel $\boldsymbol{H}(\boldsymbol{x}, \boldsymbol{x}')$. Thus, RFRR provides an approximation to kernel ridge regression that becomes increasingly accurate as the number of random features grows.

**Gaussian Universality Assumption.**    Following Simon et al. (2023), we assume that the random features can be replaced by a Gaussian projection from the RKHS associated with the deterministic kernel. Specifically, population risk is well described by the error formula obtained when the random features are replaced by $\boldsymbol{\psi}(\boldsymbol{x}) = \boldsymbol{Z}\boldsymbol{\theta}(\boldsymbol{x})$ where $[\boldsymbol{\theta}(\boldsymbol{x})]_t = \theta_t(\boldsymbol{x})$, $\boldsymbol{Z} \in \mathbb{R}^{N \times H}$ is a random Gaussian matrix with entries $Z_{ij} \sim \mathcal{N}(0, 1)$, and $H$ is the (infinite) dimensionality of the RKHS. This assumption is justified rigorously by Defilippis et al. (2024), who provide a multiplicative error bound on the resulting estimate for population risk. The stochastic kernel $\hat{\boldsymbol{H}}(\boldsymbol{x}, \boldsymbol{x}')$ can be written as the inner product of the random features:

$$\hat{\boldsymbol{H}}(\boldsymbol{x}, \boldsymbol{x}') = \frac{1}{N} \boldsymbol{\psi}(\boldsymbol{x})^\top \boldsymbol{\psi}(\boldsymbol{x}') = \frac{1}{N} \boldsymbol{\theta}(\boldsymbol{x})^\top \boldsymbol{Z}^\top \boldsymbol{Z} \boldsymbol{\theta}(\boldsymbol{x}'). \tag{5}$$

As $N \to \infty$, this stochastic kernel approaches the deterministic kernel $\boldsymbol{H}(\boldsymbol{x}, \boldsymbol{x}')$.

**RFRR Ensembles**    Ensembles of RFRR models can be constructed by averaging the predictions made by multiple independently trained RFRR models. For ensemble size $K$, we consider $\boldsymbol{\psi}^k(\boldsymbol{x}) \in \mathbb{R}^N$, $k = 1, \ldots, K$ to be the features associated with the $k^{\text{th}}$ ensemble member. The components $[\boldsymbol{\psi}^k(\boldsymbol{x})]_n = g(\boldsymbol{v}_n^k, \boldsymbol{x})$ for independently drawn $\boldsymbol{v}_n^k \sim \mu_{\boldsymbol{v}}$. The ensemble members are trained independently, and then their predictions averaged at test time:

$$f_{\text{ens}}(\boldsymbol{x}) = \frac{1}{K} \sum_{k=1}^K f^k(\boldsymbol{x}), \tag{6}$$

where $f^k(\boldsymbol{x}) = \boldsymbol{w}^{k\top} \boldsymbol{\psi}^k(\boldsymbol{x})$ is the prediction of the $k$-th random feature model. Under the assumption of Gaussian universality, we may compute theoretical learning curves by replacing $\boldsymbol{\psi}^k(\boldsymbol{x}) = \boldsymbol{Z}^k \boldsymbol{\theta}(\boldsymbol{x}) \in \mathbb{R}^{N \times H}$, where $\boldsymbol{Z}_1, \ldots, \boldsymbol{Z}_K$ are independently sampled Gaussian random matrices.

**Test Risk**    The test risk (also known as the generalization error or test error) quantifies the expected error of the learned function on unseen data. In this work, we define the test error as the mean squared error (MSE) between the predicted function $f(\boldsymbol{x})$ and the true target function $f(\boldsymbol{x})$, averaged over the data distribution $\mu_{\boldsymbol{x}}$:

$$E_g = \mathbb{E}_{\boldsymbol{x} \sim \mu_{\boldsymbol{x}}} \left[ (f(\boldsymbol{x}) - f_*(\boldsymbol{x}))^2 \right] + \sigma_\epsilon^2. \tag{7}$$

For binary classification problems, we might also consider the clasification error rate on held-out test examples under score-averaging or a majority vote (equations A.6, A.7).

## 2.2 DEGREES OF FREEDOM

Following notation similar to (Atanasov et al., 2024) and (Bach, 2023), we will write expressions in terms of the "degrees of freedom" defined as follows:

$$\text{Df}_n(\kappa) \equiv \sum_t \frac{\eta_t^n}{(\eta_t + \kappa)^n}, \qquad \text{tf}_n(\kappa) \equiv \sum_t \frac{\bar{w}_t^2 \eta_t^n}{(\eta_t + \kappa)^n}, \qquad n \in \mathbb{N}. \tag{8}$$

Intuitively, $\mathrm{Df}_n(\kappa)$ can be understood as a measures of how many modes of the kernel eigenspectrum are above a threshold $\kappa$, with the sharpness of the measurement increasing with $n$. $\mathrm{tf}_n$ is a similar measure with each mode weighted by the corresponding component of the target function.

## 3 OMNISCIENT RISK ESTIMATES FOR RANDOM FEATURE ENSEMBLES

### 3.1 THE BIAS-VARIANCE DECOMPOSITION OF $E_g$

We first review the omniscient risk estimate $E_g^1$ (superscript indicates $K = 1$) for a single RFRR model. We do not derive this well-known result here, but rather direct the reader to a wealth of derivations, including references (Atanasov et al., 2024; Canatar et al., 2021; Simon et al., 2023; Adlam & Pennington, 2020; Rocks & Mehta, 2021; Hastie et al., 2022; Zavatone-Veth et al., 2022). Translating the risk estimate into our selected notation, we may write:

$$E_g^1 \approx \frac{1}{1 - \gamma_1} \left[ -\rho \kappa_2^2 \, \mathrm{tf}_1'(\kappa_2) + (1 - \rho) \kappa_2 \, \mathrm{tf}_1(\kappa_2) + \sigma_\epsilon^2 \right] \tag{9}$$

where we have defined

$$\rho \equiv \frac{N - \mathrm{Df}_1(\kappa_2)}{N - \mathrm{Df}_2(\kappa_2)} \quad , \quad \gamma_1 \equiv \frac{1}{P} \left( (1 - \rho) \, \mathrm{Df}_1 + \rho \, \mathrm{Df}_2 \right) \tag{10}$$

and $\kappa_2$ is the solution to the following self-consistent equation:

$$\kappa_2 = \frac{\lambda N}{(P - \mathrm{Df}_1(\kappa_2))(N - \mathrm{Df}_1(\kappa_2))} \tag{11}$$

Assuming concentration of the kernel eigenfunctions, Defilippis et al. et. al. show that a dimension-free multiplicative error bound of the form $|\mathcal{E}_g^1 - E_g^1| \leq \tilde{\mathcal{O}}(N^{-1/2} + P^{-1/2}) \cdot E_g^1$, where $\mathcal{E}_g^1$ is the "true" risk and $E_g^1$ given in eq. 9, holds with high probability over the input data and random weights. We find that eq. 9 provides an accurate estimate of risk at finite $N, P$. The error formula can further be decomposed using a bias-variance decomposition with respect to the particular realization $\mathbf{Z}$ of random features:

$$E_g^1 = \mathrm{Bias}_z^2 + \mathrm{Var}_z \tag{12}$$

$$\mathrm{Bias}_z^2 \equiv \mathbb{E}_{\boldsymbol{x} \sim \mu_{\boldsymbol{x}}} \left[ \left( \mathbb{E}_{\mathbf{Z}} \left[ f(\boldsymbol{x}) \right] - f_*(\boldsymbol{x}) \right)^2 \right] + \sigma_\epsilon^2 \tag{13}$$

$$\mathrm{Var}_z \equiv \mathbb{E}_{\mathbf{Z}} \mathbb{E}_{\boldsymbol{x} \sim \mu_{\boldsymbol{x}}} \left[ \left( f(\boldsymbol{x}) - \mathbb{E}_{\mathbf{Z}} \left[ f(\boldsymbol{x}) \right] \right)^2 \right] \tag{14}$$

While the learned function $f$ in the equation above also depends on the particular realization of the dataset $\mathcal{D}$, we do not include this in the Bias-Variance decomposition because we are explicitly interested in the variance due to the realization of a finite set of random features. Furthermore, the Bias and Variance written in equations 13, 14 are expected to concentrate over $\mathcal{D}$ (Atanasov et al., 2024; Adlam & Pennington, 2020; Lin & Dobriban, 2021). Omniscient estimates for the Bias and Variance are given explicitly in (Simon et al., 2023):

$$\mathrm{Bias}_z^2 = \frac{-\kappa_2^2}{1 - \gamma_2} \, \mathrm{tf}_1'(\kappa_2) + \frac{\sigma_\epsilon^2}{1 - \gamma_2} \,, \qquad\qquad \mathrm{Var}_z = E_g^1 - \mathrm{Bias}_z^2 \,, \tag{15}$$

where $\gamma_2 \equiv \frac{1}{P} \mathrm{Df}_2(\kappa_2)$.

### 3.2 ENSEMBLING REDUCES VARIANCE OF THE LEARNED ESTIMATOR

Armed with a bias-variance decomposition of a single estimator over the realization of $\mathbf{Z}$, we can immediately write the risk estimate for an ensemble of $K$ estimators, each with an associated set of random features encapsulated by an independently drawn random Gaussian projection matrix $\mathbf{Z}^k$, $k = 1, \ldots, K$. Because the realization $\mathbf{Z}^k$ of random features is the only parameter distinguishing the ensemble members, each ensemble member will have the same expected predictor $\mathbb{E}_{\mathbf{Z}} f^k(\boldsymbol{x})$. Furthermore, because the draws of $\mathbf{Z}^k$ are independent for $k = 1, \ldots, K$, the deviations from this mean predictor will be independent across ensemble members, so that ensembling over $K$ predictors reduces the variance of the prediction by a factor of $K$:

$$E_g^K = \mathrm{Bias}_z^2 + \frac{1}{K} \, \mathrm{Var}_z \tag{16}$$

### 3.3 More is Better in Random Feature Ensembles

Predictive variance has historically been viewed as a beneficial to ensemble learning. In the case of random forests, for example, subsampling of data dimensions leads to improved performance, despite reducing the size of each decision tree ($Breiman$, 2001). In RFRR ensembles, each ensemble member is distinguished by the particular realization of its random features (i.e. the independently drawn $\boldsymbol{v}_n^k \sim \mu_{\boldsymbol{v}}$). As $N \to \infty$, the function learned by each estimator will converge to the same limiting kernel predictor. One might therefore expect *reducing* the size $N$ of each ensemble member to improve ensemble performance by increasing the diversity of the ensemble's predictors. This, however, is not the case as we prove that increasing $N$ is always beneficial to the test risk of a RFRR ensemble.

**Theorem 1.** *(More is better for RF Ensembles) Let $E_g^K(P, N, \lambda)$ denote $E_g^K$ with $P$ training samples, $N$ random features per ensemble member, ensemble size $K$, and ridge parameter $\lambda$ and any task eigenstructure $\{\eta_t\}_{t=1}^\infty$, $\{\bar{w}_t\}_{t=1}^\infty$, where $\{\eta_t\}_{t=1}^\infty$ has infinite rank. Let $K' \geq K$, $P' \geq P$ and $N' \geq N$. Then*

$$\min_\lambda E_g^{K'}(P', N', \lambda) \leq \min_\lambda E_g^K(P, N, \lambda) \tag{17}$$

*with strict inequality as long as $(K', N', P') \neq (K, N, P)$ and $\sum_t \bar{w}_t^2 \eta_t > 0$.*

*Remark* 1. In the special case $K = 1$, this reduces to the "more is better" theorem for single models proven in (Simon et al., 2023) .

Proof of this theorem follows from the omniscient risk estimate 16, and is provided in Appendix B. This theorem extends the notion that larger models, or models trained with more data, achieve better performance.

We demonstrate monotonicity with $P$ and $N$ in Fig. 1, where we plot $E_g^K$ as a function of both sample size $P$ and the network size $N$ in ensembles of ReLU random feature models applied to a binarized CIFAR-10 image classification task (see Appendix D.2). While error may increase with $P$ or $N$ at a particular ridge value $\lambda$, Error decreases monotonically provided that the ridge $\lambda$ is tuned to its optimal value. Theoretical learning curves are calculated using eq. 16, with eigenvalues $\eta_k$ and target weights $\bar{w}_k$ determined by computing the NNGP kernel corresponding to the infinite-feature limit of the ReLU RF model (see Appendix D.3 for details). Numerically, we verify that error monotonicity with $P$ and $N$ holds at the level of a 0-1 loss on the predicted classes of held-out test examples for both score-averaging and majority-vote ensembling over the predictors (see fig. S3).

We compare ridge-optimized error across ensemble sizes $K$ in fig. S1, finding that increasing sample size $P$ and network size $N$ are usually more effective than ensembling over multiple networks in reducing predictor error, indicating that for the binarized CIFAR-10 RFRR classification task, bias is the dominant contribution to error. Similarly, ensembling over multiple networks gives meager improvements in performance relative to increasing network size in deep feature-learning ensembles Vyas et al. (2023).

## 4 No Free Lunch from Random Feature Ensembles

It is immediate from eq. 16 that increasing the size of an ensemble reduces the error. However, with a fixed memory capacity, a machine learning practitioner is faced with the decision of whether to train a single large model, or to train an ensemble of smaller models and average their predictions. Here, we prove a "no free lunch" theorem which says that, given a fixed total number of features $M$ divided evenly among $K$ random feature models, then the lowest possible risk will always be achieved by $K = 1$, provided that the ridge is tuned to its optimal value. Furthermore, this ridge-optimized error increases monotonically with $K$.

**Theorem 2.** *(No Free Lunch From Random Feature Ensembles) Let $E_g^K(P, N, \lambda)$ denote $E_g^K$ with $P$ training samples, $N$ random features per ensemble member, ridge parameter $\lambda$, ensemble size $K$, and task eigenstructure $\{\eta_t\}_{t=1}^\infty$, $\{\bar{w}_t\}_{t=1}^\infty$, where $\{\eta_t\}_{t=1}^\infty$ has infinite rank. Let $K' < K$. Then*

$$\min_\lambda E_g^{K'}(P, M/K', \lambda) \leq \min_\lambda E_g^K(P, M/K, \lambda) \tag{18}$$

*with strict inequality as long as $\sum_t \bar{w}_t^2 \eta_t > 0$.*

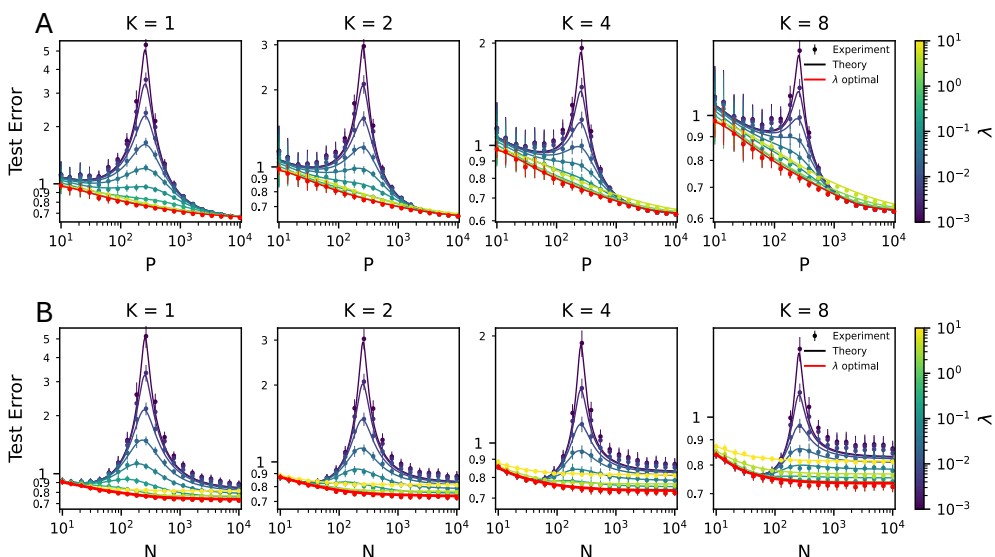

Figure 1: "More is better" in random feature ensembles. We perform ReLU RFRR on a binarized CIFAR-10 classification task and compare the empirical test risk to the omniscient risk estimate (eq. 16). (A) We fix $N = 256$ and vary both $P$ and $K$. Color corresponds to the regularization $\lambda$. Markers show numerical experiments and dotted lines theoretical predictions. Error is monotonically decreasing with $P$ provided that the regularization $\lambda$ is tuned to its optimal value. (B) Same as (A) except that $P = 256$ is fixed and $K$, $N$ are varied. Markers and error bars show mean and standard deviation over 50 trials.

The proof follows a similar strategy to the proof of theorem 1, and is provided in appendix B. We test this prediction by performing ensembled ReLU RFRR on the binarized CIFAR-10 classification task in figure 2. We find that increasing $K$ while keeping the total number of features $M$ fixed always degrades the optimal test risk. The strength of this effect, however, depends on the size of the training set. For larger training sets ($P \gg N$), the width of each ensemble member becomes the constraining factor in each predictor's ability to recover the target function. However, when $P \ll N$, the optimal loss is primarily determined by $P$, so that optimal error only begins increasing appreciably with $K$ once $N = M/K \lesssim P$ (fig. 2 B). We again find similar behavior of the 0-1 loss under score-average and majority-vote ensembling (see fig. S6). While the ridge-optimized error is always minimal for $K = 1$, we notice in fig. 2C that near-optimal performance can be obtained with $K > 1$ over a wider range of $\lambda$ values, suggesting that ensembling may offer improved robustness in situations where fine-tuning of the ridge parameter is not possible. Further robustness benefits have been reported for regression ensembles of heterogeneous size (Ruben & Pehlevan, 2023).

Theorems 1 and 2 together guarantee that a larger ensemble of smaller RFRR ensembles can only outperform a smaller ensemble of larger RFRR models when the total parameter count of the former exceeds that of the latter. We formalize this fact in the following corollary:

**Corollary 1.** *Let $E_g^K(N)$ be the test risk of an ensemble of $K$ RFRR models each with $N$ features given by eq. 16. Suppose $K' > K$ or $N' < N$. It follows from Theorems 1 and 2 that*

$$\min_\lambda E_g^{K'}(N') \leq \min_\lambda E_g^K(N) \Rightarrow K'N' \geq KN. \tag{19}$$

We demonstrate this result on synthetic tasks with power-law structure and on the binarized CIFAR-10 classification task in fig. S2. For ensembles in the over-parameterized regime ($N \gg P$), this bound appears to be tight.

## 5 WIDTH-BOTTLENECKED SCALING OF RANDOM FEATURE ENSEMBLES

To gain a better understanding of the trade-off between ensemble size $K$ and total feature count $M$, we ask how the error $E_g^K$ scales with total model size $M$ under the standard "source" and "capacity"

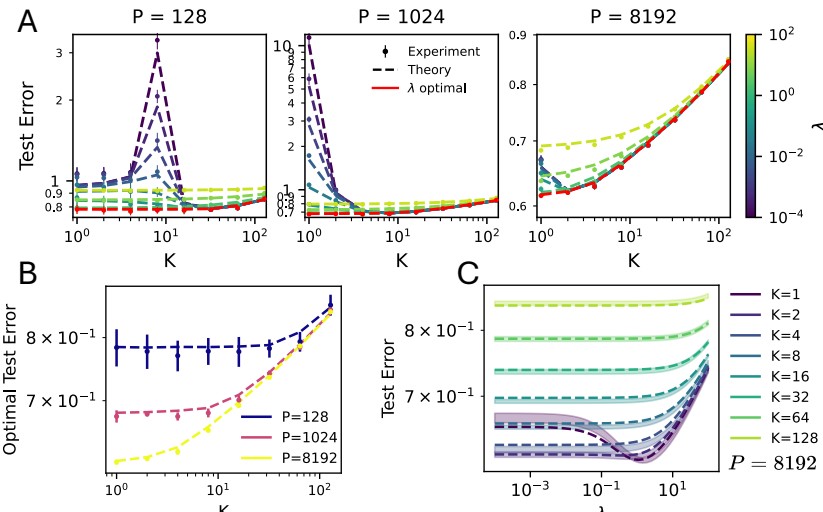

Figure 2: No Free Lunch from Random Feature Ensembles. We perform kernel RF regression on a binarized CIFAR 10 classification task. (A) We vary $K$ and $N$ while keeping total parameter count $M = 1024$ fixed. The sample size $P$ is indicated above each plot. (B) Error $E_g^K$ optimized over the ridge parameter $\lambda$ increases monotonically with $K$ provided the total parameter count $M$ is fixed. Dashed lines show theoretical prediction using eq. 16 and markers and error-bars show mean and standard deviation of the risk measured in numerical simulations across 10 trials. (C) We show error as a function of $\lambda$ for each $K$ value simulated and $P = 8192$. Dashed lines show theoretical prediction using eq. 16 and shaded regions show standard deviation of risk measured in numerical simulations across 10 trials.

constraints on the task eigenstructure (Cui et al., 2023; Caponnetto & De Vito, 2006; Bordelon et al., 2020; Defilippis et al., 2024). In particular, we assume that the kernel eigenspectrum decays as $\eta_t \sim t^{-\alpha}$ with $\alpha > 1$ and the target's power in each mode decays as $\bar{w}_t^2 \eta_t \sim t^{-(1+2\alpha r)}$. We also assume that $N = M/K \ll P$, so that we are in the width-bottlenecked regime (otherwise, error scaling is dominated by the sample size $P$) (Bahri et al., 2024; Maloney et al., 2022; Bordelon et al., 2024a; Atanasov et al., 2024). To understand the scaling of $E_g^K$ with $M$, we introduce a "growth exponent" $\ell \in [0, 1]$ which controls the joint scaling of $K$ and $N$ with $M$:

$$\ell \in [0, 1] \qquad K \sim M^{1-\ell} \qquad N \sim M^\ell, \tag{20}$$

so that when $\ell = 0$ the ensemble grows with $M$ by adding additional ensemble members of a fixed size, and when $\ell = 1$ the ensemble grows by adding parameters to a fixed number of networks. Under these conditions, we find:

$$E_g^K \sim M^{-s}, \qquad s = \min\left(2\alpha\ell\min(r, 1), 1 - \ell + 2\alpha\ell\min\left(r, \frac{1}{2}\right)\right), \tag{21}$$

with $s = 2\alpha\ell\min(r, 1)$ corresponding to the scaling of the bias and $s = 1 - \ell + 2\alpha\ell\min(r, 1/2)$ corresponding the scaling of the variance (reduced by a factor of $1/K$). A full derivation is provided in appendix C. These results reify the "no free lunch" result, as the optimal scaling law is always achieved when $\ell = 1$. For difficult tasks, defined as having $r < 1/2$, bias always dominates the error scaling and the scaling exponent increases linearly with $\ell$. However, when $r > 1/2$, there will be a certain value $\ell^*$ above which error scaling is dominated by the variance term. When $r > 1/2$, the scaling exponent of the variance increases from 1 to $\alpha$ over the range $\ell \in [0, 1]$. If $\alpha \gtrsim 1$, this can approach a flat line, and the dependence of the scaling exponent on $\ell$ can become weak, so that *near-optimal* scaling can be achieved for any $\ell > \ell^*$. When $1/2 < r < 1$, this transition occurs at $\ell^* = 1/(1 + \alpha(2r - 1))$ and when $r > 1$ it occurs at $\ell^* = 1/(1 + \alpha)$.

We plot these scaling laws with $\alpha = 1.5$ in the regimes where $r < 1/2$, $1/2 < r < 1$, and $r > 1$ in fig. 3, along with the results of numerical simulations of linear RF regression on synthetic Gaussian

datasets. As anticipated, for difficult tasks, where $r < 1/2$, the scaling law improves linearly with $\ell$. However, for easier tasks ($r > 1/2$), we see that, a near-optimal scaling law can be achieved as long as $\ell > \ell^*$ (fig. 3, center and right columns).

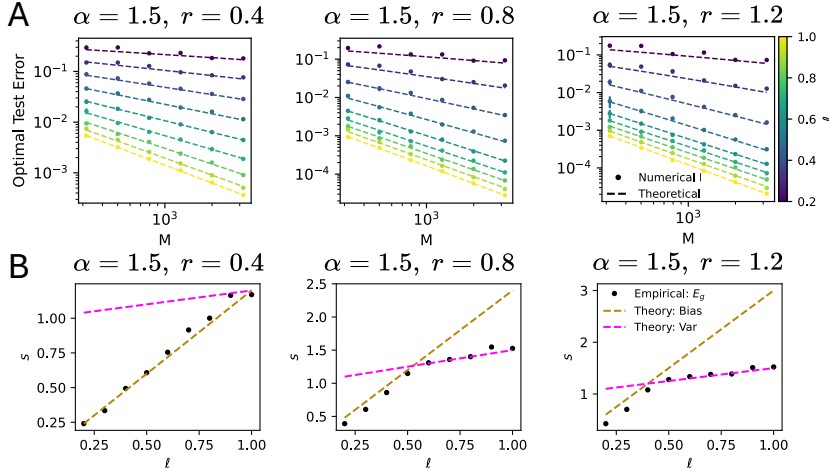

Figure 3: width-bottlenecked scaling laws of kernel RF regression under source and capacity constraints. We fix $P = 15,000$, $\alpha = 1.5$, and $r \in \{0.4, 0.8, 1.2\}$ and calculate $E_g^K$ as a function of $M$ with $N = M^\ell$ and $K = M^{(1-\ell)}$ using both the omniscient risk estimate (eq. 16) and numerical simulation of a linear Gaussian random-feature model (eq. A.12). (A) Plots of $E_g^K$ vs. $M$ at different $\ell$ values reveal that $\ell$ controls the scaling law of the error. (B) We plot the theoretical scaling exponents (eq. 21): Bias $\sim 2\alpha\ell \min(r, 1)$, Var $\sim 1 - \ell + 2\alpha\ell \min(r, \frac{1}{2})$ along with the scaling laws obtained by fitting the risks obtained by numerical simulation.

We also determine the scaling behavior of the $\mathrm{ReLU}$ RFRR ensembles on the binarized CIFAR-10 and MNIST classification tasks. For both tasks, we calculate the statistics of the limiting kernel eigenspectrum $\{\eta_t\}_{t=1}^\infty$ and target weights $\{\bar{w}_t\}_{t=1}^\infty$ and fit their spectral decays to the source and capacity constraints. We find that for CIFAR-10, $\alpha \approx 1.33, r \approx 0.038$ and for MNIST $\alpha \approx 1.46, r \approx 0.14$, which places both tasks squarely in the difficult regime with $r < 1/2$. In figure 4, we show that the predicted scaling exponents of eq. 21.

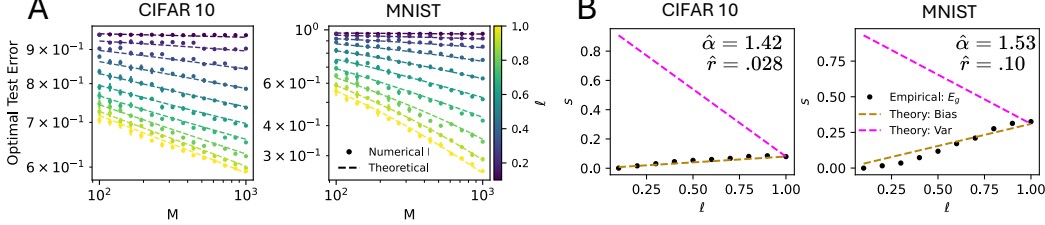

Figure 4: Scaling laws provide a good description of width-bottlenecked RFRR ensembles.(A) we plot error as a function of $M$ at optimal ridge value for $\mathrm{ReLU}$ random-feature models applied to the binarized CIFAR-10 (left) and MNIST (right) classification tasks. (B) We plot theoretically predicted scaling exponents (eq. 21) for the bias and variance contributions to risk, as well as empirical power-law fits to risk in numerical simulations of RFRR models (see Appendix D.3, fig. S7)

## 6 NO FREE LUNCH FROM FEATURE LEARNING ENSEMBLES

Here, ask whether the "no free lunch from ensembles" principle proven for RFRR carries over to ensembles of deep neural networks. In the lazy training regime, deep neural networks reduce to

kernel machines, with random features given by the gradients of the loss at initialization (Chizat et al., 2019). Consequently, random feature models are reliable toy models for lazy training, with the number of random features as a proxy for the number of parameters in the network (Jacot et al., 2018; Lee et al., 2019; Chizat et al., 2019; Bordelon et al., 2020; Canatar et al., 2021). For example, RFRR exhibits overfitting at finite width and sample size Atanasov et al. (2022). Feature learning can, however, complicate the relationship between network size and performance, if the strength of feature-learning depends on network width, as in NTK parameterization (Jacot et al., 2018; Aitchison, 2020). To make "fair" comparisons between large models and ensembles of smaller models, we seek instead a parameterization which keeps training dynamics consistent across widths, with monotonic improvements in performance as width increases.

Maximal update parameterization ($\mu P$) (Geiger et al., 2020; Mei et al., 2018; Rotskoff & Vanden-Eijnden, 2022; Yang & Hu, 2021; Bordelon & Pehlevan, 2022) accomplishes this desired width-consistency (Vyas et al., 2023). $\mu P$ is the unique parameterization in which the network's infinite width limit converges and permits feature learning in finite time (Yang et al., 2022; Bordelon & Pehlevan, 2022). $\mu P$ is similar to the NTK parameterization, except we center and scale the output of the neural network inversely with a richness parameter (Chizat et al., 2019):

$$\tilde{f}(x;\theta) = \frac{1}{\gamma}\left(f(x;\theta) - f(x;\theta_0)\right), \qquad \gamma = \gamma_0\sqrt{N}, \quad \eta = \eta_0\gamma^2 \tag{22}$$

so that the richness $\gamma_0$ and learning rate $\eta_0$ are constants and $\gamma$ and $\eta$ scale with network size (Geiger et al., 2020; Mei et al., 2018; Rotskoff & Vanden-Eijnden, 2022; Yang & Hu, 2021; Bordelon & Pehlevan, 2022). At small $\gamma$, small changes in the weights are sufficient to interpolate the training data, yielding a model well-approximated as a linear model with the kernel given by the NTK at initialization. This is known as lazy learning. At large $\gamma$, large weight updates are necessary to change the network's output, and the model learns task-relevant features. This is known as rich learning.

We train ensembles of deep convolutional neural networks (CNNs) on the CIFAR-10 image classification task, sweeping over ensemble size $K$, richness $\gamma$, and weight decay $\lambda$. We use a small CNN architecture with two CNN layers (figs. 5A, S8), as well as a larger ResNet18 architecture (figs. 5A, S9). The width of the convolutional and MLP layers are varied with $K$ to keep the total parameter count fixed (details in Appendix E). In the "lazy" regime ($\gamma \ll 1$), we find that accuracy decreases monotonically with $K$, provided weight decay is tuned to its optimal value. And, while at some intermediate values of $\gamma$ error may increase with $K$, monotonicity is restored when weight decay and richness $\gamma$ are jointly tuned to their optimal values (figs. 5A,B, S8, S9).

We also test the performance of ensembles of transformers trained on the C4 language modeling task. We train in the online setting where each sample is used no more than once. No weight decay is used. In agreement with results from (Vyas et al., 2023), we find that across richness parameters $\gamma$, error is monotonically increasing with $K$ (fig. 5B).

To summarize, our findings suggest that "no free lunch from ensembles" holds for deep ensembles trained with $\mu P$ parameterization under any of following conditions:

- In the lazy training regime ($\gamma \to 0$) when the weight decay is tuned to its optimal value.
- When weight decay and richness $\gamma$ are jointly tuned to their optimal values.
- When richness $\gamma$ is fixed and training is performed *online* (i.e. without repeating data).

# 7 DISCUSSION

An important limitation of our work is the assumption of statistically homogeneous ensembles. We consider each ensemble member to be trained on the same dataset, and to perform the same task. However, successes have been achieved using ensembles with *functional specialization*, where different sub-networks are trained on different datasets to perform different sub-tasks relevant to the overall goal of the ensemble. For example, mixture of experts (MoE) models (Jacobs et al., 1991; Lepikhin et al., 2020; Fedus et al., 2022) might offer a way to cleverly scale model size using ensembles that outperforms the scaling laws for single large networks. We leave a theory of ensembled regression which allows for functional specialization as an objective for future work.

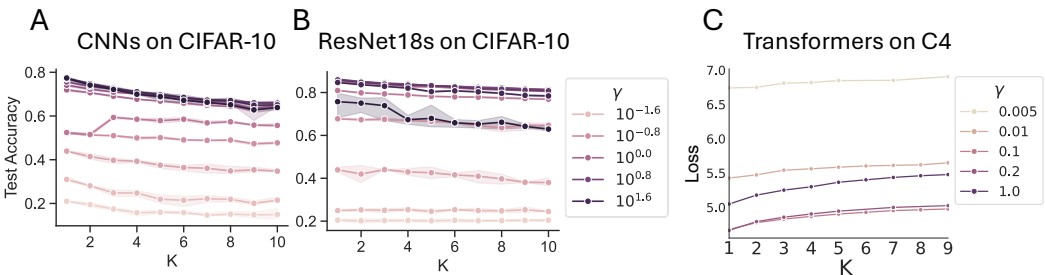

Figure 5: (A, B) No Free Lunch from deep CNN ensembles on CIFAR-10. At optimal weight decay, performance decays (decrease in test accuracy) monotonically with ensemble size $K$ when the total number of parameters $M$ is fixed, for lazy training ($\gamma \ll 1$) and at optimal richness. We test a small CNN architecture with 2 convolutional layers and one MLP layer (A) and ResNet18 ensembles (B). (C) No Free Lunch from Transformer ensembles trained online for 5000 steps on the C4 dataset. For all $\gamma$, the performance decays (indicated by an increase in loss) monotonically with the ensemble size.

Another limitation of our work is the absence of feature-learning in the RFRR toy model, which prevents a direct application of our theory to deep ensembles in the rich regime. Nevertheless, we find that when deep networks are trained using $\mu$P parameterization, the "no free lunch from ensembles" principle holds empirically provided the weight decay and richness parameter $\gamma$ are tuned to their optimal values. This fact might be proven rigorously by extending a recent analytical model of feature-learning networks to the ensembled case (Bordelon et al., 2024b).

Our study also connects to recent work on scaling laws in deep learning (Kaplan et al., 2020; Hoffmann et al., 2022; Bordelon et al., 2024a;b), which observe that the test error of neural networks tends to improve predictably as a power-law with the number of parameters and the size of the dataset used during training. With our scaling-law analysis, we extend the power-laws predicted using random-feature models (Bahri et al., 2024; Maloney et al., 2022; Bordelon et al., 2020; 2024a; Defilippis et al., 2024) to the case where model size is scaled up by jointly increasing the ensemble size $K$ and parameters per ensemble member $N$ according to a "growth exponent" $\ell$ defined in eq. 20. While optimal scaling is always achieved by fixing $K$ and scaling up network size, for sufficiently easy tasks ($r > 1/2$), near-optimal scaling laws can be achieved by growing both network size and ensemble size, provided network size grows *quickly enough* with total parameter count. Because feature-learning networks can dynamically align their representations to the target function Bordelon et al. (2024b), the scaling laws for deep ensembles may be dramatically improved by feature-learning effects.

## 8 CONCLUSION

In this work, we analyzed a trade-off between ensemble size and features-per-ensemble-member in the tractable setting of RFRR. We prove a "no free lunch" theorem which states that optimal performance is always achieved by allocating all features to a single, large RFRR model, provided that the ridge parameter is tuned to its optimal value. A scaling-laws analysis reveals that the sharpness of this trade-off depends sensitively on the structure of the task. In particular, near-optimal scaling laws can be achieved by RFRR ensembles, provided the task is sufficiently aligned with the top modes of the limiting kernel eigenspectrum. We confirm that in deep neural network ensembles with fixed total parameter count, increasing ensemble size $K$ leads to worse performance in both a computer vision and language modeling task, provided that both the weight decay *and* the richness parameters are tuned to their optimal values. In addition to explaining the general trend from massively ensembled predictors to large models with jointly trained parameters in recent years, our results have practical implications for model design and resource allocation in real-world settings, where model size is limited.

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

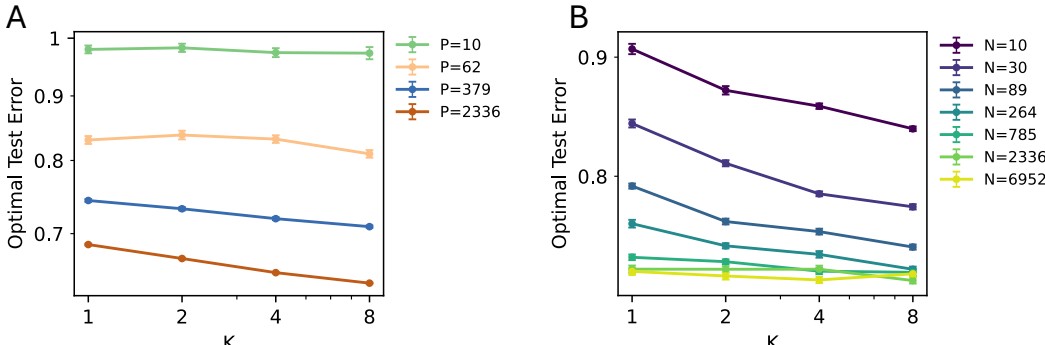

Figure S1: $E_g^k$ at optimal ridge as a function of ensemble size $K$ for binarized CIFAR-10 RFRR classification. (A) We fix $N = 256$, $P$ values indicated in legend. (B) We fix $P = 256$, $N$ values indicated in legend. Error bars show standard deviation across 10 trials.

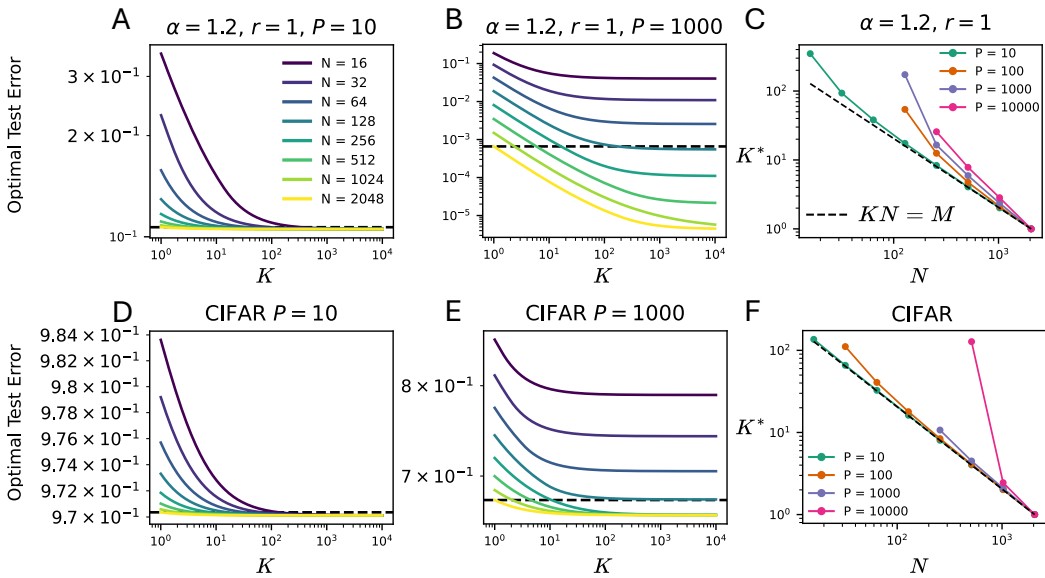

Figure S2: (A, B, D, E) We plot theoretical values for $E_g^k$ at optimal ridge as a function of ensemble size $K$ for RFRR with power-law eigenstructure with source exponent $r = 1$ and capacity $\alpha = 1.2$ (A, B) and for the NNGP kernel associated with the binarized CIFAR-10 classification task (D, E) . Random features per ensemble member $N$ shown in the legend. The dotted black line shows $E_g^1$ for a single RFRR model with $N = M = 2048$ features. Sample size $P$ is indicated in the title. (C, F) We plot the ensemble size $K^*$ for which an ensemble of RFRR models with size $N$ performs at least as well as single RFRR model with $M = 2048$ random features. $P$ values indicated in legend. As predicted by Theorem 2, all curves lie above the dotted line $KN = M$. This bound appears tight when $P \ll N$.

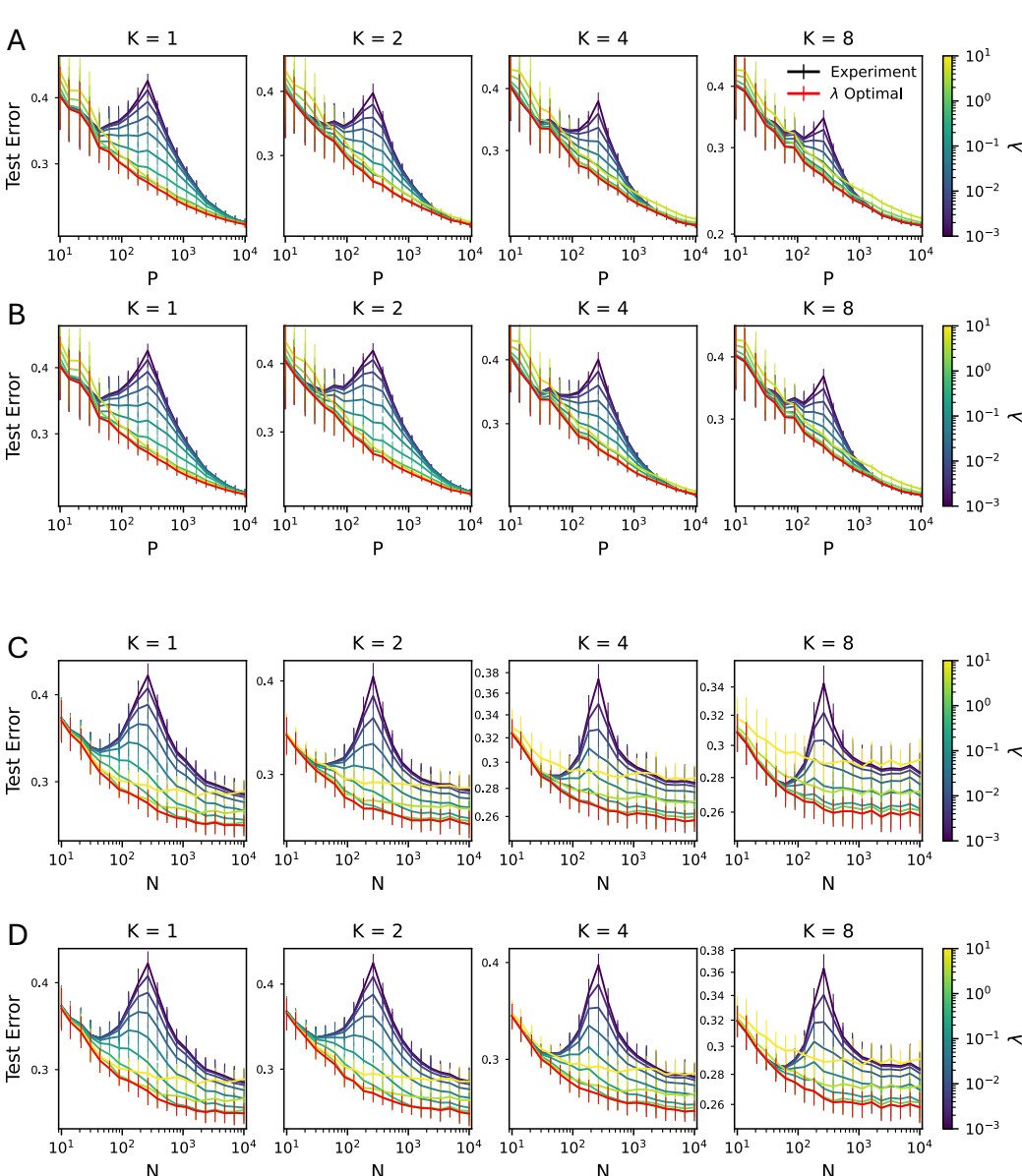

Figure S3: 0-1 Loss for binarized CIFAR-10 RFRR task under score-averaging (A, C) and majority vote (B, D) ensembling schemes. As in Fig. 1, Errors are shown (A, B) as a function of $P$ for fixed $N = 256$ and (B) as a function of $N$ for fixed $P = 256$. $K$ value indicated in title and $\lambda$ value in colorbar. Red line indicates optimal ridge determined by grid search. Markers and error bars show mean and standard deviation over $50$ trials.

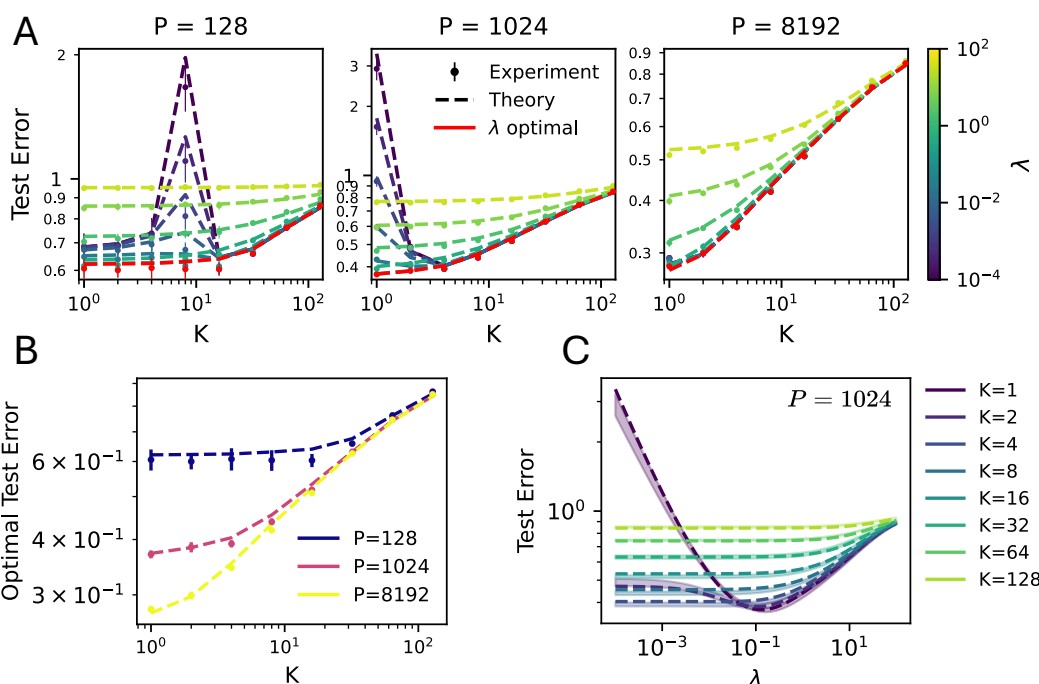

Figure S4: No Free Lunch from Ensembles of Random Feature Models. $E_g$ for kernel RF regression on an MNIST classification task. (A) Warying $K$ and $N$ while keeping total parameter count $M = 1024$ fixed. The sample size $P$ is indicated above each plot. (B) Error $E_g^K$ optimized over the ridge parameter $\lambda$ increases monotonically with $K$ provided the total parameter count $M$ is fixed. Dashed lines show theoretical prediction using eq. 16 and markers and error-bars show mean and standard deviation of the risk measured in numerical simulations across 10 trials. (C) We show error as a function of $\lambda$ for each $K$ value simulated and $P = 8192$. Dashed lines show theoretical prediction using eq. 16 and shaded regions show standard deviation of risk measured in numerical simulations across 10 trials.

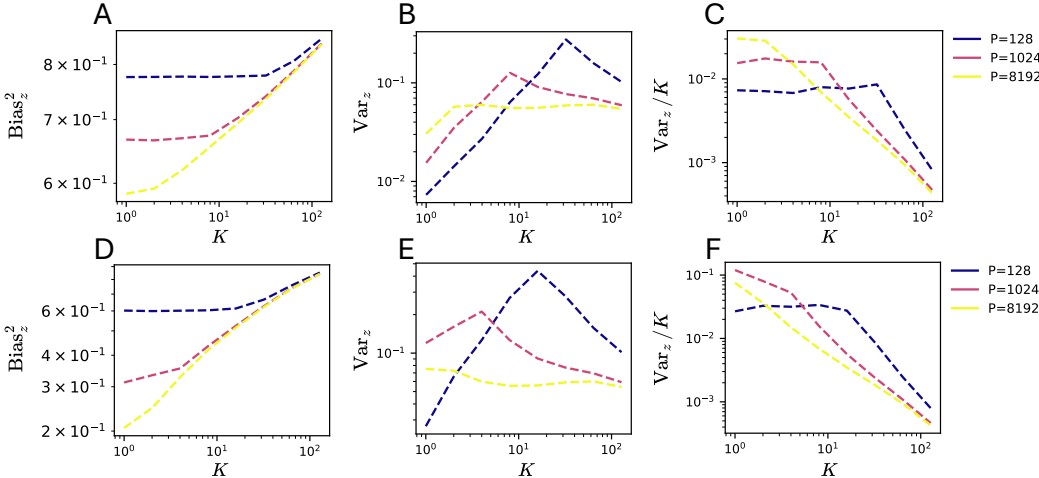

Figure S5: Bias-Variance decomposition of error at optimal ridge for binarized CIFAR-10 (A, B, C) and MNIST (D, E, F) RFRR tasks. We vary $K$ and $N$ while keeping total parameter $M = 1024$ fixed. $\text{Bias}_z^2$ (A, D), single-predictor variance $\text{Var}_z$ (B, E), and ensemble-predictor variance $\text{Var}_z^2/K$ (C, F) are calculated from theoretical expressions 15.

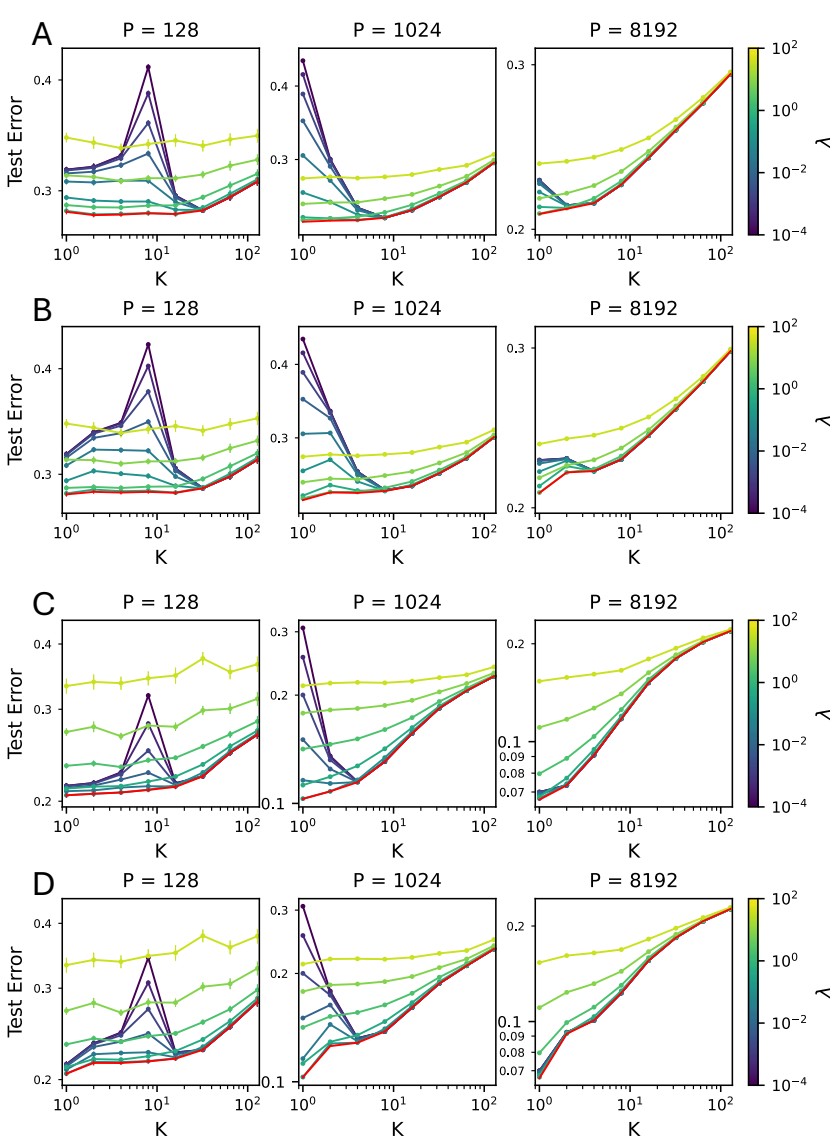

Figure S6: 0-1 loss for binarized CIFAR-10 (A, B) and MNIST (C, D) RFRR classification tasks under score-average (A, C) and majority vote (B, D) ensembling. We sweep $K$ and $N$ keeping $M = KN = 1024$ fixed. Sample size $P$ indicated in titles. Colorbar indicates ridge parameter. Red indicates optimal ridge determined by grid search.

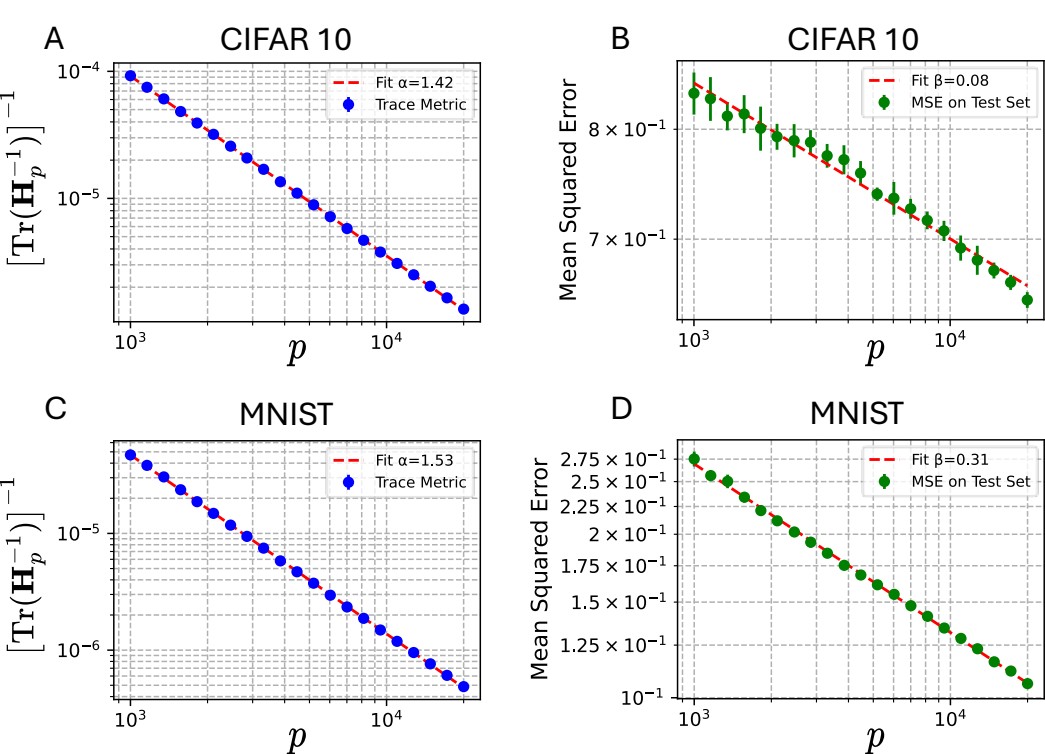

Figure S7: We measure the eigenspectrum of the NNGP kernel applied to the CIFAR-10 and MNIST datasets, as well as the target weights for the binarized classification tasks described in Appendix D.2. Estimates for the source and capacity exponents are obtained by fitting the "trace metric" $\left[\mathrm{tr}\left[\boldsymbol{H}_p\right]^{-1}\right]^{-1}$ and the MSE loss of kernel ridge regression with the limiting NNGP kernel to power laws (see Appendix D.4 for details).

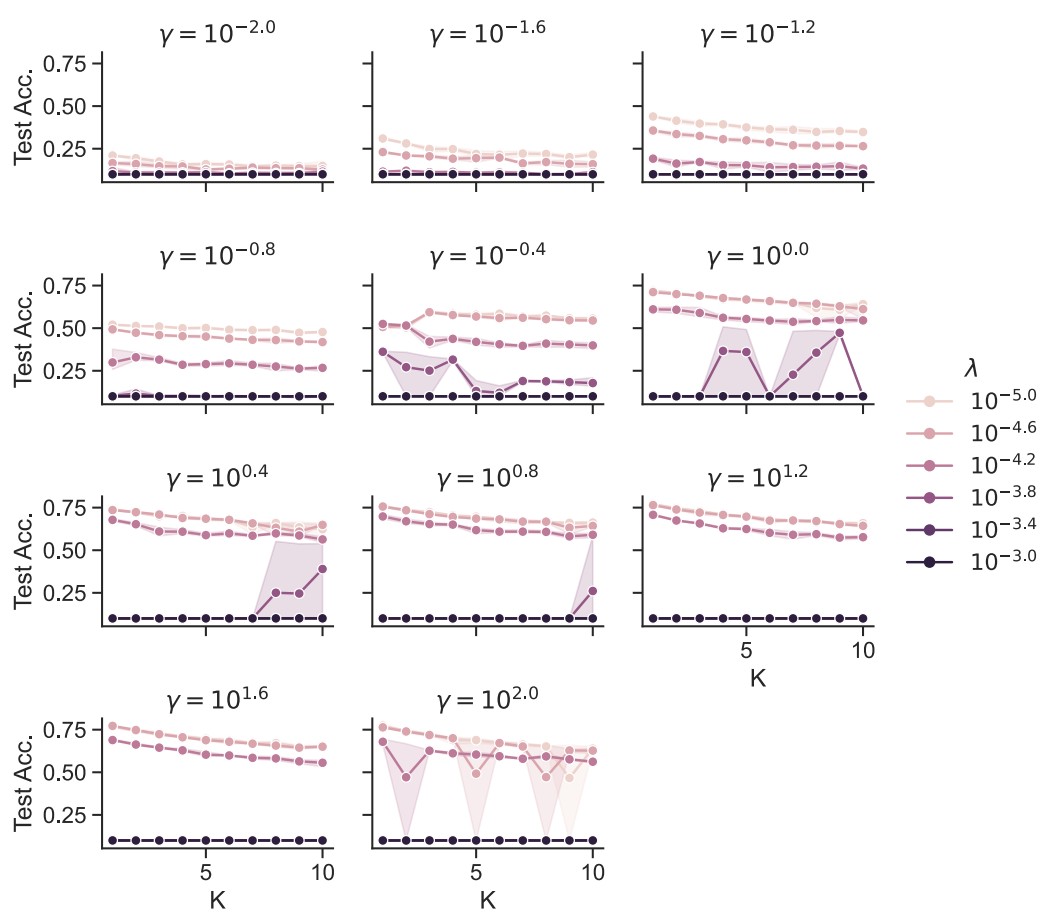

Figure S8: CNNs on CIFAR-10 for varying richness $\gamma$ and weight decay $\lambda$. Total parameter count is held fixed while ensemble size $K$ is varied.

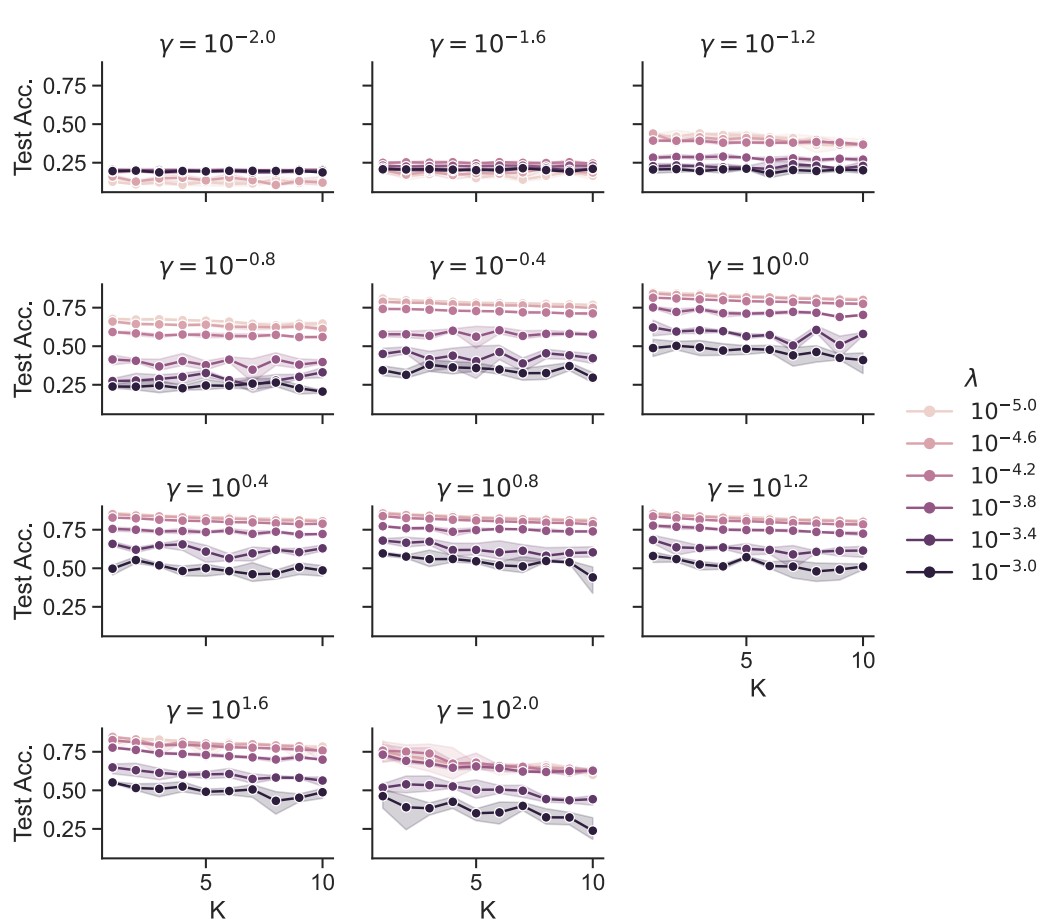

Figure S9: ResNet18s on CIFAR-10 for varying richness $\gamma$ and weight decay $\lambda$. Total parameter count is held fixed while ensemble size $K$ is varied.

# A EXTENDING THE RANDOM-FEATURE EIGENFRAMEWORK TO ENSEMBLES

We consider the RFRR setting as described by (Simon et al., 2023), extended to ensembles of predictors. A detailed description of this extended framework is Let $\mathcal{D} = \{\boldsymbol{x}_p, y_p\}_{i=1}^P$ be a training set of $P$ examples, where $\boldsymbol{x}_p \in \mathbb{R}^D$ are input features and $y_p \in \mathbb{R}$ are target values generated by a noisy ground-truth function $y_p = f_*(\boldsymbol{x}_p) + \epsilon_p$ and the label noise $\epsilon_p \overset{\text{i.i.d.}}{\sim} \mathcal{N}(0, \sigma_\epsilon^2)$.

We consider an ensemble of $K$ random feature models, each with $N$ features. The total number of features is thus $M = K \cdot N$. For each model $k = 1, \ldots, K$, we sample $N$ feature vectors $\{\boldsymbol{w}_n^k\}_{n=1}^N$ i.i.d. from a measure $\mu_{\boldsymbol{v}}$ over $\mathbb{R}^B$ (we will use upper indices to represent the index of the ensemble member, and lower indices to represent indices of the training examples and features). An ensemble of $K$ *featurization transformations* are defined as $\boldsymbol{\psi}^k : \boldsymbol{x} \mapsto (g(\boldsymbol{v}_n^k, \boldsymbol{x}))_{n=1}^N$ where $g : \mathbb{R}^B \times \mathbb{R}^D \to \mathbb{R}$ is square-integrable with respect to $\mu_{\boldsymbol{x}}$ and $\mu_{\boldsymbol{v}}$. The predictions of the ridge regression models are then given as $f^k(\boldsymbol{x}) = \boldsymbol{w}^k \cdot \boldsymbol{\psi}(\boldsymbol{x})$, where the weight vectors $\boldsymbol{w}^k$ are determined by standard linear ridge regression with a ridge parameter $\lambda$:

$$\hat{\boldsymbol{w}}^k = \left(\frac{1}{N}\boldsymbol{\Psi}^{k\top}\boldsymbol{\Psi}^k + \lambda\mathbf{I}\right)^{-1}\frac{\boldsymbol{\Psi}^{k\top}\boldsymbol{y}}{N} \tag{A.1}$$

Where the matrices $\boldsymbol{\Psi}^k \in \mathbb{R}^{N \times P}$ have columns $[\boldsymbol{\psi}^k(\boldsymbol{x}_1), \cdots, \boldsymbol{\psi}^k(\boldsymbol{x}_P)]$ and the vector $\boldsymbol{y} \in \mathbb{R}^P$ has $[\boldsymbol{y}]_p = y_p$. For each ensemble member, this is equivalent to the kernel ridge regression predictor:

$$f^k(\boldsymbol{x}) = \hat{\boldsymbol{h}}_{x,\mathcal{X}}\left(\hat{\boldsymbol{H}}_{\mathcal{X}\mathcal{X}} + \lambda\boldsymbol{I}_N\right)^{-1}\boldsymbol{y} \tag{A.2}$$

Where the matrix $[\hat{\boldsymbol{H}}_{\mathcal{X}\mathcal{X}}]_{pp'} = \hat{\boldsymbol{H}}^k(\boldsymbol{x}_p, \boldsymbol{x}_{p'})$ and the vector $[\hat{\boldsymbol{h}}_{x,\mathcal{X}}]_p = \hat{\boldsymbol{H}}^k(\boldsymbol{x}, \boldsymbol{x}_{\boldsymbol{p}})$ with the stochastic finite-feature kernel

$$\hat{\boldsymbol{H}}^k(\boldsymbol{x}, \boldsymbol{x}') = \frac{1}{N}\sum_{n=1}^N g(\boldsymbol{w}_n^k, \boldsymbol{x})g(\boldsymbol{w}_n^k, \boldsymbol{x}') \qquad k = 1, \ldots, K \tag{A.3}$$

In the limit of infinite features, this stochastic kernel converges to a deterministic limit $\hat{\boldsymbol{H}}^k(\boldsymbol{x}, \boldsymbol{x}') \to \boldsymbol{H}(\boldsymbol{x}, \boldsymbol{x}')$. Because we consider the feature function $g$ to be shared across ensemble members, this limit is independent of $k$. The ensemble prediction is the average of individual model predictions:

$$f_{\text{ens}}(\boldsymbol{x}) = \frac{1}{K}\sum_{k=1}^K f^k(\boldsymbol{x}) \tag{A.4}$$

Finally, we measure the test error as the mean-squared error of the ensemble as the mean-squared error on a held out-test sample:

$$E_g \equiv \mathbb{E}_{\boldsymbol{x} \sim \mu_{\boldsymbol{x}}}\left[(f_{\text{ens}}(\boldsymbol{x}) - f_*(\boldsymbol{x}))^2\right] + \sigma_\epsilon^2 \tag{A.5}$$

For binary classification problems, we may be more interested in the classification error rate for the learned predictor. Given an ensemble of scalar output "scores" $f^1(\boldsymbol{x}), \ldots, f^K\boldsymbol{x}$, two possible schemes to assign the class of the test example $\boldsymbol{x}$ are score-averaging and majority-vote ensembling (Loureiro et al., 2022):

$$f_{\text{ens}}^{\text{SA}}(\boldsymbol{x}) = \text{Sign}\left(\sum_{k=1}^K f^k(\boldsymbol{x})\right) \qquad\qquad \text{(Score-Average)} \tag{A.6}$$

$$f_{\text{ens}}^{\text{MV}}(\boldsymbol{x}) = \text{Sign}\left(\sum_{k=1}^K \text{Sign}\left(f^k(\boldsymbol{x})\right)\right) \qquad\qquad \text{(Majority-Vote)} \tag{A.7}$$

The classification error rate is then given as the probability of mislabeling a held-out test example.

## A.1 SPECTRAL DECOMPOSITION OF THE KERNEL

The feature function $g$ permits a spectral decomposition as follows:. Let $T : L^2(\mu_{\boldsymbol{v}}) \to L^2(\mu_{\boldsymbol{x}})$ be the linear operator defined by:

$$(Tr)(x) = \int_{\mathbb{R}^B} r(\boldsymbol{v})g(\boldsymbol{v}, \boldsymbol{x})d\mu_{\boldsymbol{v}}(\boldsymbol{v}) \tag{A.8}$$

The singular value decomposition of $T$ ((Kato, 1966)) yields orthonormal bases $\{\zeta_n\}_{n=1}^{\infty}$ of $\text{Ker}^{\perp}(T) \subset L^2(\mu_v)$ and $\{\phi_n\}_{n=1}^{\infty}$ of $L^2(\mu_n)$, where $\{\eta_t\}_{t=1}^{\infty}$ are the eigenvalues (in decreasing order) and $\{\zeta_t\}_{t=1}^{\infty}$ the corresponding eigenvectors integral operator $\Sigma : L^2(\mu_x) \to L^2(\mu_x)$ given by

$$(\Sigma u)(\boldsymbol{x}) = \int_{\mathbb{R}^D} u(\boldsymbol{x}')\boldsymbol{H}(\boldsymbol{x}',\boldsymbol{x})d\mu_{\boldsymbol{x}}(\boldsymbol{x}') \tag{A.9}$$

We can write $\Sigma = TT^{\star}$, where $T^{\star} : L^2(\mu_x) \mapsto L^2(\mu_v)$ denotes the adjoint of $T$. The feature function $g$ can then be decomposed as $g(\boldsymbol{v},\boldsymbol{x}) = \sum_{t=1}^{\infty} \sqrt{\eta_t}\zeta_t(\boldsymbol{v})\phi_t(\boldsymbol{x})$.

Under these conditions, we may write the stochastic finite-feature kernel functions as:

$$\hat{\boldsymbol{H}}^k(\boldsymbol{x},\boldsymbol{x}') = \frac{1}{N}\sum_{n=1}^{N}\sum_{t,t'=1}^{\infty}\sqrt{\eta_t\eta_{t'}}\zeta_t(\boldsymbol{v}_n^k)\zeta_{t'}(\boldsymbol{v}_n^k)\phi_t(\boldsymbol{x})\phi_{t'}'(\boldsymbol{x}') \tag{A.10}$$

Using the orthonormality of the bases, the deterministic limit of the kernel function can then be expanded as

$$\boldsymbol{H}(\boldsymbol{x},\boldsymbol{x}') = \sum_t \eta_t \phi_t(\boldsymbol{x})\phi_t(\boldsymbol{x}') = \sum_t \theta_t(\boldsymbol{x})\theta_t(\boldsymbol{x}') \tag{A.11}$$

where we have defined $\theta_t(\boldsymbol{x}) \equiv \sqrt{\eta_t}\phi_t(\boldsymbol{x})$. We see that that the singular values $\{\eta_t\}$ of the operator $T$ double as the eigenvalues of the limiting kernel operator $\boldsymbol{H}$. We will assume that the ground truth function $f_*(\boldsymbol{x})$ can then be decomposed as: $f_*(\boldsymbol{x}) = \sum_t \bar{w}_t \theta_t(\boldsymbol{x})$. Any component of $f_*$ which does not lie in the RKHS of the kernel could, in principle, be absorbed into the noise $\sigma_{\epsilon}^2$ (Canatar et al., 2021).

## A.2 Gaussian Universality Ansatz and the connection to Linear Random Feature Ridge Regression

As in (Simon et al., 2023; Atanasov et al., 2022), we adopt the Gaussian universality ansatz, which states that the expected train and test errors are unchanged if we replace $\{\zeta_t\}$ and $\{\phi_t\}$ with random Gaussian functions $\{\tilde{\zeta}_t\}$ and $\{\tilde{\phi}_t\}$ such that $\tilde{\zeta}_t(\boldsymbol{v}) \sim \mathcal{N}(0,1)$ and $\tilde{\phi}_t(\boldsymbol{x}) \sim \mathcal{N}(0,1)$ for $\boldsymbol{v} \sim \mu_{\boldsymbol{v}}$ and $\boldsymbol{x} \sim \mu_{\boldsymbol{x}}$, respectively.

The finite-feature stochastic kernels can then be written $\hat{\boldsymbol{H}}^k(\boldsymbol{x},\boldsymbol{x}') = \tilde{\boldsymbol{\theta}}^k(\boldsymbol{x}) \cdot \tilde{\boldsymbol{\theta}}^k(\boldsymbol{x}')$ where $\tilde{\boldsymbol{\theta}}^k(\boldsymbol{x}) \equiv \boldsymbol{Z}^k\boldsymbol{\theta}(\boldsymbol{x})$ and the entries of $\boldsymbol{Z}^k \in \mathbb{R}^{N \times H}$ are drawn i.i.d. as $\mathcal{N}(0,1/N)$ and we have defined $H$ to be the (possibly infinite) dimensionality of the reproducing kernel Hilbert space (RKHS) of $\boldsymbol{H}$. The learned functions can then be written as

$$f^k(\boldsymbol{x}) = \hat{\boldsymbol{w}}^k \cdot \boldsymbol{\psi}(\boldsymbol{x}) \quad, \quad \boldsymbol{w}^k = \boldsymbol{Z}^{k\top}\left(\boldsymbol{Z}^k\boldsymbol{\Theta}^{\top}\boldsymbol{\Theta}\boldsymbol{Z}^{k\top} + \lambda\boldsymbol{I}\right)^{-1}\boldsymbol{Z}^k\boldsymbol{\Theta}^{\top}\boldsymbol{y} \tag{A.12}$$

Where $\boldsymbol{\Theta} = [\boldsymbol{\theta}(\boldsymbol{x}_1),\cdots,\boldsymbol{\theta}(\boldsymbol{x}_P)]$, and the vectors $\boldsymbol{\theta}(\boldsymbol{x}_p) \sim \mathcal{N}(\boldsymbol{0},\boldsymbol{\Lambda})$, with $\boldsymbol{\Lambda}$ a diagonal matrix with $\boldsymbol{\lambda}_{tt} = \eta_t$. This is precisely the setting of a *linear* random-feature model with *data* covariance spectrum $\{\eta_1,\eta_2,...\}$ (Atanasov et al., 2024). Under the Gaussian universality ansatz, we can therefore re-cast RFRR as linear RFRR with the role of the spectrum of the "data" played by the spectrum $\{\eta_t\}_{t=1}^{\infty}$ of the limiting deterministic kernel $\boldsymbol{H}$.

# B Proof of Theorems 1 and 2

In this section, we will refer to the condition that $\sum_t \bar{w}_t^2 \eta_t > 0$ as the task having a "learnable component." It can be shown from eq. 11 that $\text{Df}_1 \leq \min(N,P)$. Because $N$ and $P$ are finite and $\text{rank}(\{\eta_t\}_{t=1}^{\infty})$ is infinite, it follows that $\kappa_2 > 0$. The following inequalities then hold strictly:

$$\text{Df}_2 < \text{Df}_2 \qquad\qquad \gamma_2 < \gamma_1 \qquad\qquad 0 < \rho < 1 \tag{B.1}$$

Furthermore, the inequality $\text{tf}_2(\kappa) \leq \text{tf}_1(\kappa)$ holds, with strict inequality when the target has a learnable component.

## B.1 "BIGGER IS BETTER" THEOREMS

We begin by proving theorem 1. We note that it suffices to prove that $E_g^K$ decreases monotonically with $K$, $N$, and $P$ separately, since any transformation $(K, N, P) \rightarrow (K', N', P')$ can be taken in steps $K \rightarrow K'$, $N \rightarrow N'$, $P \rightarrow P'$.

**Monotonicity with $K$** The fact that when $K' > K$ and all other variables are held fixed, $E_g^K$ decreases is immediately evident from the form of eq. 16, because $\mathrm{Bias}_z^2$ and $\mathrm{Var}_z$ are independent of $K$. Furthermore, the inequality is strict as long as $\mathrm{Var}_z > 0$, which is valid as long as the task has a learnable component.

**Monotonicity with $P$** Consider a transformation $P \rightarrow P'$ with $P' > P$. Examining eq. 11, we see that it is always possible to increase the ridge $\lambda$ such that $\kappa_2$ remains fixed. We then rewrite $E_g^K$ as:

$$E_g^K = (1 - \frac{1}{K}) \, \mathrm{Bias}_z^2 + \frac{1}{K} E_g^1 \tag{B.2}$$

With $\kappa_2$ and $N$ fixed, we see that only the prefactor of $\frac{1}{1-\gamma_2}$ in $E_g^1$ will be affected, so that $E_g^1$ decreases with $P$. Note also that $\gamma_1$ is a decreasing function of $P$. With $\kappa_2$ and $N$ fixed, it follows that $E_g^1$ decreases with $P$. Finally, because $K$ is fixed, $E_g^K$ will decrease with $P$.

**Monotonicity with $N$** Consider a transformation $N \rightarrow N'$ with $N' > N$. Examining eq. 11, we see that it is always possible to increase the ridge $\lambda$ so that $\kappa_2$ remains constant. With $\kappa_2$, $P$ fixed, $\mathrm{Bias}_z^2$ is fixed as well. From eq. B.2, it then suffices to show that $E_g^1$ decreases. To see this, recall that $\rho$ is an increasing function of $N$, and $\gamma_1$ is a decreasing function of $\rho$. We then have that $\gamma_1$ is a decreasing function of $N$, so that the prefactor of $\frac{1}{1-\gamma_1}$ in eq. 9 is decreasing with $N$.

## B.2 "NO FREE LUNCH" FROM ENSEMBLES THEOREM

We now prove theorem 2. We first recall the form of the error to be:

$$E_g^K = \mathrm{Bias}_z^2 + \frac{1}{K} \left( E_g^1 - \mathrm{Bias}_z^2 \right) \tag{B.3}$$

We define the variable $\nu \equiv \frac{1}{K}$. By analytical continuation, it suffices to show that test risk decreases as $\nu$ increases. Rewriting the self-consistent equation in terms of $\nu$, we have;

$$\kappa_2 = \frac{\lambda N}{(P - \mathrm{Df}_1(\kappa_2))(\nu M - \mathrm{Df}_1(\kappa_2))} \tag{B.4}$$

Consider a transformation $\nu \rightarrow \nu'$ where $\nu' > \nu$. We se that it is always possible to increase $\lambda$ so that $\kappa_2$ remains fixed. Note that $\mathrm{Bias}_z^2$ depends only on $\kappa_2$ and $P$, so that as $\nu$ (and therefore $N$) vary, $\mathrm{Bias}_z^2$ remains fixed. From eq. 16, it therefore suffices to show that $\nu(E_g^1 - \mathrm{Bias}_z^2)$ decreases with $\nu$. Rearranging terms, we have:

$$\nu(E_g^1 - \mathrm{Bias}_z^2) = \nu \left[ \frac{-\rho \kappa_2^2 \, \mathrm{tf}_1'(\kappa_2) + (1-\rho)\kappa_2 \, \mathrm{tf}_1(\kappa_2)}{1 - \gamma_1} - \frac{-\kappa_2^2 \, \mathrm{tf}_1'(\kappa_2)}{1 - \gamma_2} \right] \tag{B.5}$$

$$+ \nu \left[ \frac{1}{1 - \gamma_1} - \frac{1}{1 - \gamma_2} \right] \sigma_\epsilon^2 \tag{B.6}$$

We first show that

$$\frac{d}{d\nu} \left[ \nu \left( \frac{1}{1 - \gamma_1} - \frac{1}{1 - \gamma_2} \right) \right] < 0, \tag{B.7}$$

To see this, recall that $\rho = (\nu M - \mathrm{Df}_1)/(\nu M - \mathrm{Df}_2)$. Because $\mathrm{Df}_1 > \mathrm{Df}_2$, this is a monotonically increasing function of $\nu$. We may write $\gamma_1 = \frac{1}{P}\left[(1-\rho)\,\mathrm{Df}_1 + \rho\,\mathrm{Df}_2\right]$. From this equation it is clear that $\gamma_1 > \gamma_2$. Differentiating with respect to $\nu$, we get

$$\frac{d\rho}{d\nu} = M\frac{(\mathrm{Df}_1 - \mathrm{Df}_2)}{(\nu M - \mathrm{Df}_2)^2} \tag{B.8}$$

$$\frac{d\gamma_2}{d\nu} = -\frac{1}{P}(\mathrm{Df}_1 - \mathrm{Df}_2)\frac{d\rho}{d\nu} = -\frac{M}{P}\frac{(\mathrm{Df}_1 - \mathrm{Df}_2)^2}{(\nu M - \mathrm{Df}_2)^2} \tag{B.9}$$

Using these, we have:

$$\frac{d}{d\nu}\left[\nu\left(\frac{1}{1-\gamma_1} - \frac{1}{1-\gamma_2}\right)\right] \tag{B.10}$$

$$= \left(\frac{1}{1-\gamma_1} - \frac{1}{1-\gamma_2}\right) + \frac{\nu\frac{d\gamma_1}{d\nu}}{(1-\gamma_1)^2} \tag{B.11}$$

$$= \frac{(\mathrm{Df}_1 - \mathrm{Df}_2)^2}{(1-\gamma_1)(\nu M - \mathrm{Df}_2)}\left[\frac{1}{P(1-\gamma_2)} - \frac{M\nu}{P(1-\gamma_1)(\nu M - \mathrm{Df}_2)}\right] \tag{B.12}$$

$$< 0 \tag{B.13}$$

where in the last line, we have used the facts that $\gamma_1 > \gamma_2$ and $\mathrm{Df}_2 \leq \nu M$. To show that

$$\frac{d}{d\nu}\left[\nu\left(\frac{-\rho\kappa_2^2\,\mathrm{tf}_1'(\kappa_2) + (1-\rho)\kappa_2\,\mathrm{tf}_1(\kappa_2)}{1-\gamma_1} - \frac{-\kappa_2^2\,\mathrm{tf}_1'(\kappa_2)}{1-\gamma_2}\right)\right] \leq 0, \tag{B.14}$$

we first note that $-\rho\kappa_2^2\,\mathrm{tf}_1'(\kappa_2) + (1-\rho)\kappa_2\,\mathrm{tf}_1(\kappa_2)$ can be equivalently written as $-\kappa_2^2\,\mathrm{tf}_1'(\kappa_2) + (1-\rho)\kappa_2\,\mathrm{tf}_2(\kappa_2)$. The above derivative can then be broken into two parts:

$$-\kappa_2^2\,\mathrm{tf}_1'\frac{d}{d\nu}\left[\nu\left(\frac{1}{1-\gamma_1} - \frac{1}{1-\gamma_2}\right)\right] + \kappa_2\,\mathrm{tf}_2\frac{d}{d\nu}\left[\frac{\nu(1-\rho)}{1-\gamma_1}\right] \tag{B.15}$$

We have already shown that the derivative in the first term is negative. Furthermore, $-\kappa_2^2\,\mathrm{tf}_1' \geq 0$, with strict equality holding when the task has a learnable component. To see that the derivative in the second term is negative, note that $\rho$ is an increasing function of $\nu$. Because $\gamma_1$ is a decreasing function of $\rho$, $\gamma_1$ is therefore an decreasing function of $\nu$. The denominator $1 - \gamma_1$ inside the derivative increases with $\nu$. Furthermore, the numerator $\nu(1-\rho)$ can be written as $(\mathrm{Df}_1 - \mathrm{Df}_2)\frac{\nu}{\nu M - \mathrm{Df}_2}$. With $\kappa_2$ fixed (so that $\mathrm{Df}_1 - \mathrm{Df}_2 > 0$ is fixed), this is a strictly decreasing function of $\nu$. It follows that

$$\kappa_2\,\mathrm{tf}_2\frac{d}{d\nu}\left[\frac{\nu(1-\rho)}{1-\gamma_1}\right] \leq 0, \tag{B.16}$$

with strict inequality holding as long as $\mathrm{tf}_2 > 0$, which is true whenever the task has a learnable component.

## C    DERIVATION OF SCALING LAWS

In this section, we derive the width-bottlenecked scaling laws given in section 5, using methods described in (Atanasov et al., 2024). We assume that the kernel eigenspectrum decays as $\eta_t \sim t^{-\alpha}$ and the power of the target function in the modes decays as $\bar{w}_t^2\eta_t \sim t^{-(1+2\alpha r)}$, and examine the regime where $P \gg N$, so that the width of the ensemble members is the bottleneck to signal recovery. We begin by analyzing the self-consistent equation for $\kappa_2$, reproduced here for clarity:

$$\kappa_2 = \frac{\lambda N}{(P - \mathrm{Df}_1(\kappa_2)(N - \mathrm{Df}_1(\kappa_2)))} \tag{C.1}$$

Because $\mathrm{Df}_1(\kappa_2) < \min(N, P)$ and $N \ll P$, it follows that $P \gg \mathrm{Df}_1(\kappa_2)$. We can therefore approximate the fixed-point equation as

$$P\kappa_2 \approx \frac{\lambda}{1 - \frac{1}{N}\mathrm{Df}_1(\kappa_2))} \tag{C.2}$$

We approximate $\mathrm{Df}_1$ using an integral:

$$\mathrm{Df}_1(\kappa_2) \approx \int_1^\infty \frac{t^{-\alpha}}{t^{-\alpha} + \kappa_2} dt \tag{C.3}$$

Making the change of variables $u = t\kappa_2^{1/\alpha}$, we get

$$\mathrm{Df}_1(\kappa_2) \approx \kappa_2^{-1/\alpha} \int_{\kappa_2^{1/\alpha}}^\infty \frac{du}{1 + u^\alpha} \tag{C.4}$$

Plugging back into the fixed-point equation, we arrive at

$$\kappa_2 P \approx \frac{\lambda}{1 - \frac{\kappa_2^{-1/\alpha}}{N}\int_{\kappa_2^{1/\alpha}}^\infty \frac{du}{1+u^\alpha}} \tag{C.5}$$

Next, we make the ansatz that $\kappa_2 \sim N^{-q}$. The fixed point equation becomes

$$PN^{-q} \sim \frac{\lambda}{1 - N^{\frac{q}{\alpha}-1}\int_{N^{-q/\alpha}}^\infty \frac{du}{1+u^\alpha}} \tag{C.6}$$

The left size of this equation will be very large, due to the separation of scales $P \gg N$. The only feasible way for the right side to scale with $P$ is for the denominator to become very small as $N$ grows. This is only possible if $N^{q/\alpha} \sim N$, so that $q = \alpha$. We therefore have $\kappa_2 \sim N^{-\alpha}$.

With this scaling for $\kappa_2$, we have $\mathrm{Df}_1, \mathrm{Df}_2 \sim N$. It is then clear that $\gamma_2 \to 0$ for $P \gg N$. Furthermore, because $\rho \in [0, 1]$, $\gamma_1 \to 0$ for $P \gg N$. The prefactors of $\frac{1}{1-\gamma_2}$ and $\frac{1}{1-\gamma_1}$ can therefore be ignored.

We may then write

$$\kappa_2\, \mathrm{tf}_1(\kappa_2) \sim \int_1^\infty \frac{t^{-(1+2\alpha r)}}{1 + t^{-\alpha}/\kappa_2} dt \sim N^{-2\alpha r}\int_{1/N}^\infty \frac{u^{-(1+2\alpha r)}}{1 + u^{-\alpha}} du \tag{C.7}$$

where $u = t\kappa^{1/\alpha}$ and we have made the substitution $\kappa \sim N^{-\alpha}$. We get two contributions to the integral: when $u$ is near $1/N$, we get a contribution (including the prefactor) which scales as $N^{-\alpha}$. When $u$ is away from $1/N$, the integral contributes a constant factor and we get a contribution that scales as the prefactor $N^{-2\alpha r}$.

Similarly, we may write:

$$-\kappa_2^2\, \mathrm{tf}_1'(\kappa_2) \sim \int_1^\infty \frac{t^{-(1+2\alpha r)}}{(1 + t^{-\alpha}/\kappa_2)^2} dt \sim N^{-2\alpha r}\int_{1/N}^\infty \frac{u^{-(1+2\alpha r)}}{(1 + u^{-\alpha})^2} du \tag{C.8}$$

The contributions from the component of the integral near $1/N$ will now scale as $N^{-2\alpha}$, and the contribution away from $1/N$ will remain $N^{-2\alpha r}$. Combining these results, we arrive at separate scaling laws for the bias and variance terms of the error:

$$\mathrm{Bias}_z^2 \sim N^{-2\alpha \min(r,1)} \tag{C.9}$$

$$\mathrm{Var}_z^2 \sim N^{-2\alpha \min(r,\frac{1}{2})} \tag{C.10}$$

Finally, to obtain eq. 21, we put $N \sim M^\ell$ and $K \sim M^{1-\ell}$ and substitute into eq. 16. We find that, in terms of $M$, the bias and variance scale as:

$$\text{Bias}_z^2 \sim M^{-2\ell\alpha\min(r,1)} \tag{C.11}$$

$$\frac{1}{K}\text{Var}_z \sim M^{-\left(1-\ell+2\alpha\ell\min\left(r,\frac{1}{2}\right)\right)} \tag{C.12}$$

The scaling of the total loss for an ensemble will be dominated by the more slowly-decaying of these two terms.

## C.1 SAMPLE-BOTTLENECKED SCALING

We next examine the case where $P \ll N$. Here, we find that $\text{Df}_1(\kappa_2), \text{Df}_2(\kappa_2) \ll N$ so that $\rho \to 1$, $\text{Var}_z \to 0$. The only significant contribution to the error will come from $\text{Bias}_z^2$, which will scale as (Atanasov et al., 2024):

$$E_g^K \sim P^{-2\alpha\min(r,1)} \qquad (\lambda \ll P^{1-\alpha}) \tag{C.13}$$

$$E_g^K \sim (\lambda/P)^{2\min(r,1)} \qquad (\lambda \gg P^{1-\alpha}) \tag{C.14}$$

We therefore see that the ensemble size $K$ and network size $N$ have no effect on the scaling law in $P$ provided $P \ll N$.

# D   RFRR ON REAL DATASETS

## D.1   NUMERICAL EXPERIMENTS WITH SYNTHETIC GAUSSIAN DATA

For a given value of $\alpha$ and $r$, we fix a large value $D \gg P, M$ and generate eigenvalues $\eta_t \propto t^{-\alpha}$ and target weights $\bar{w}_t \sim t^{-\frac{1}{2}(1-\alpha+2\alpha r)}$. The $\eta_t$ are normalized so that $\sum_t \eta_t = 1$ and $\sum_t \bar{w}_t^2 \eta_t = 1$. We generate random features as $\boldsymbol{\Theta} = \boldsymbol{\Sigma}\boldsymbol{X} \in \mathbb{R}^{D\times P}$, where $X_{ij} \sim \mathcal{N}(0,1)$ and $\boldsymbol{\Sigma}$ is the diagonal matrix with entry $\Sigma_{tt} = \eta_t$. Labels are assigned as $\boldsymbol{y} = \boldsymbol{\Psi}^\top\bar{\boldsymbol{w}} \in \mathbb{R}^P$. For each $k = 1,\ldots,K$, we perform linear RFRR (eq. A.12) with an independently drawn projection matrix $\boldsymbol{Z}^k$ with entries drawn from $\mathcal{N}(0,1/N)$. The prediction of the ensemble is then given as the mean over the $K$ learned predictors.

## D.2   NUMERICAL EXPERIMENTS ON BINARIZED MNIST AND CIFAR-10 WITH RELU FEATURES

We perform RFRR on CIFAR-10 and MNIST datasets. To construct the dataset, we sort the images into two classes. For CIFAR-10, we assign a label $y = +1$ to images of "things one could ride" together (airplane, automobile, horse, ship, truck) and a label $y = -1$ for "things one ought not to ride" (bird, cat, deer, dog, frog) (Simon et al., 2023). For MNIST, we assign a label $y = +1$ to digits $0-4$, and a label $-1$ to digits $5-9$. We construct $K$ feature maps as $\psi(^k\boldsymbol{x}) = \frac{1}{\sqrt{N}}\text{ReLU}\left(\boldsymbol{V}^{k\top}\boldsymbol{x}\right)$, where $V_{ij}^k \sim \mathcal{N}(0,2/D)$. Here, $D$ is the data dimensionality ($D = 3072$ for CIFAR-10 and 784 for MNIST). Then, for each ensemble member $k = 1,\ldots,K$, we train a linear regression model on the features $\psi^k$. In the infinite-feature limit, the finite-feature kernels will converge to the "NNGP kernel" for a single-hidden-layer Relu network (*Lee et al.*, 2018).

## D.3   THEORETICAL PREDICTIONS

We evaluate the omniscient risk estimate 16 numerically using vectors storing the values of $\{\eta_t\}_{t=1}^\infty$ and $\{\bar{w}_t\}_{t=1}^\infty$. In the case of synthetic data, these vectors are readily available. For the MNIST and CIFAR-10 tasks, we approximate these vectors by evaluating the infinite-width neural network gaussian process (NNGP) kernel using the neural tangents library (Novak et al., 2019). For 30,000 images from the training sets of both MNIST and CIFAR 10, we evaluate the kernel matrix $[\boldsymbol{H}]_{p,p'} = \boldsymbol{H}(\boldsymbol{x}_p, \boldsymbol{x}_{p'})$, and diagonalize the kernel matrix to determine the eigenvectors and eigenvalues. To be precise, with $P = 30,000$, we calculate the eigenvalues $\{\eta_1, \eta_2, \ldots, \eta_P\}$ and eigenvectors $\{\boldsymbol{u}_1, \ldots, \boldsymbol{u}_P\}$ of the sample-normalized kernel matrix $\frac{1}{P}\boldsymbol{H}$. We then assign the weights of the target function as $\bar{w}_t = \frac{1}{\sqrt{P\eta_t}}\boldsymbol{u}_t^\top\boldsymbol{y}$, where $\boldsymbol{y} \in \mathbb{R}^P$ is the vector of labels associated to our $P$ samples.

We then solved the self-consistent equation (eq. 11) using the Bisection method of the scipy library (Virtanen et al., 2020), and evaluate eq. 16 to determine the predicted risk.

### D.4 MEASURING POWER LAW EXPONENTS OF KERNEL RIDGE REGRESSION TASKS

In fig. 4B, we plot theoretical predictions for the scaling exponents of the bias and variance contributions to error for the binarized CIFAR-10 and MNIST classification tasks. To calculate these exponents, we need access to the "ground truth" source and capacity exponents $\hat{\alpha}$ and $\hat{r}$ characterizing the kernel eigenstructure of the dataset and the target function, such that the eigenvalues $\{\eta_1, \eta_2, \dots\}$ of the NNGP kernel decay as $\eta_t \sim t^{-\hat{\alpha}}$ and the weights of the target function $\{\bar{w}_1, \bar{w}_2, \dots\}$ decay as $\eta_t \bar{w}_t^2 \sim t^{-(1+2\hat{\alpha}\hat{r})}$. To estimate $\hat{\alpha}$, we calculate the "trace metric" $\left[\text{tr}\left[\boldsymbol{H}_p^{-1}\right]\right]^{-1}$ $\hat{\alpha}$ is then obtained by fitting to the relationship $\left[\text{tr}\left[\boldsymbol{H}_p^{-1}\right]\right]^{-1} \sim p^{-\alpha}$, where $\boldsymbol{H}_p \in \mathbb{R}^{p \times p}$ is the empirical NNGP kernel for $p$ randomly drawn samples from the dataset (Wei et al., 2022; Simon et al., 2023) (figs. S7.A,C). To estimate the source exponent $r$, we use the scaling law for Kernel Ridge Regression (with the limiting infinite-feature NNGP kernel) which dictates that for small ridge, $E_g \sim p^{-2\alpha \min(r,1)}$ (see Appendix C.1). Following Simon et al., we fit the MSE loss for Kernel Ridge Regression with the limiting NNGP kernel to a power law decay $E_g \sim p^{-\beta}$ (figs. S7.B,D), and assign $\hat{r} = \beta/2/\hat{\alpha}$.

## E DEEP ENSEMBLE EXPERIMENTAL DETAILS

### E.1 NATURAL LANGUAGE PROCESSING TASKS

The transformers used for the WikiText2 experiments are implemented in the maximal update parameterization following the setup in (Bordelon et al., 2024c). We fix depth $L = 6$, number of heads $H = 12$, and vary width $N$ such that $k\sqrt{N} \approx 60$ to keep the parameter count between ensembles approximately the same since parameters scale quadratically with width. The transformer is composed of alternating blocks of causal multiheaded attention and a 2-layer MLP. We make sure to center the models' outputs and vary feature learning in the models by dividing the output by $\gamma = \gamma_0\sqrt{N}$.

### E.2 CNN RESULTS ON CIFAR-10

The CNNs used for the CIFAR-10 experiments consist of two CNN layers and an MLP layer. To compare ensembles of different sizes fairly while keeping the total number of parameters fixed, we adopt a fixed ratio between layer sizes in each CNN. If the first CNN layer has width (channels) $c$, then the second CNN layer has width $2c$, and the MLP has width $5c$. The value of $c$ is varied such that the total ensemble maintains approximately the same number of parameters $M$ across varying numbers of ensemble members $K$.

We employ standard data augmentation techniques during training. All training images are randomly cropped, resized, and flipped horizontally. Our models are trained using SGD with base learning rate 0.1 and early stopping, with a patience of five epochs.

