# OpenReview forum: "No Free Lunch from Random Feature Ensembles"
_ICLR.cc/2025/Conference — Submitted to ICLR 2025_

### Official Review · Reviewer_6nYe · 2024-10-29

**Soundness:** 1
**Presentation:** 2
**Contribution:** 1
**Rating:** 3
**Confidence:** 4

**Summary:**

This paper analyzes the ensemble of K ridge regressions on random features size M/K and one right regression on random features size M. And concludes that the later one is always the optimal under the optimal L2 regularization. This work also use some toy experiments (e.g. two to three layers CNN) to compare the ensemble of K small models and one large models.

**Strengths:**

- The writing is clear. The paper is easy to read.

**Weaknesses:**

- In this paper, Text, Theory and Experiments are not consistent.
In text, this paper assumes a fixed "computational budget" (line 256), which is resonable, then links this assumption to "a fixed total number of features"  (line 257).  However, a fixed computational budget is not necessary leads to a fixed total number of features.

The theory discusses the ensemble case of regression on random features. Concludes that one right regression on random features size M is always the optimal under the **optimal L2 regularization**, compared with the ensemble of K ridge regressions on random features size M/K. However, that is trivial. Section 9.1 in [1] shows a dropout on regression equals to an ensemble of expensional number of small regressiors, equals to L2 regularization. Since **an ensemble of expensional number of small regressiors** is not worse that **M/K regressors** (same as the Eq 16 in this work), an "optimal L2 regularization" is of course not worse than "M/K regressors"

The cifar experiments in this work does not link to the theory, i.e. without neither regression nor random features.



[1] Srivastava, N., Hinton, G., Krizhevsky, A., Sutskever, I., & Salakhutdinov, R. (2014). Dropout: a simple way to prevent neural networks from overfitting. The journal of machine learning research, 15(1), 1929-1958.





typos:
line 141  f(x)

**Questions:**

- The CNN experiment on CIFAR10 contains two CNN layers and one linear layer. That is already too small for CIFAR10 tasks. Reducing the size of this small network would further hurt the performance. Does the same phenomena in this paper hold on larger networks?  [2] shows the ensemble multiple small networks outperforms a large network.


[2] Zhang, J., & Bottou, L. (2023, July). Learning useful representations for shifting tasks and distributions. In International Conference on Machine Learning (pp. 40830-40850). PMLR.

---

> ### Author Response · Authors · 2024-11-21
> **Response to Reviewer Comments (Part 1)**
>
> Thank you for your review.  We will respond in two parts due to character limits.
>
> In this paper, Text, Theory and Experiments ... number of features.
> > **Response**: Thank you for this comment, which we agree with.  We have now clarified in the main text that we consider constraints on total parameter count.  The performance of a single large model and ensembles of smaller models are usually compared with reference to the total parameter count, including in [2].  Total parameter count determines the memory required to store a learned model, which is a significant constraint in practice.
>
> The theory discusses ... not worse than "M/K regressors"
>
> > **Response**: We thank the reviewer for calling this result to our attention. However, we believe our result is stronger.
>
> > Before giving our reasons for why our result is stronger, we are also aware of results showing equivalence between a single ridge-regularized regressor with access to the full inputs and an infinite ensemble of unregularized, subsampling or "sketched" regressors [3, 4].  An analogous (and true!) statement in the setting we describe would read that an infinite ensemble of random-feature ridge regresssion models with zero ridge is equivalent to the limiting (i.e. infinite-feature) kernel ridge regression model.  It would follow that even infinitely large ensembles of random feature models could never outperform the limiting kernel regression model with optimal ridge.  This corresponds to the limit $M, K \to \infty$ in Theorem 2.
>
> > However, the result we present is stronger because it applies at finite $M$.  Rather than comparing a single regressor with access to the full (infinite) set of kernel features to an ensemble of subsampling regressors, we compare a regressor with access to a single large random projection of the kernel features to an ensemble of regressors with access to smaller random projections of the kernel features.  Because of this distinction, we find that ensembles with $K > M/N$ can actually *outperform* a single regressor of size $M$. To demonstrate this, we plot error as a function of $K$ ensembles of RFRR models with $N$ features each in a new supplemental fig. S2.  We see that an ensemble of RFRR models with $N$ features each can outperform a single RFRR model of size $M>N$ only if $K>K^*$.  Theorems 1 and 2 together guarantee that the value $K^*$ required to achieve better performanced with the ensemble is at least as large as $M/N$.  Similarly, if we fix $K$ and allow $N$ to grow, the value $N^*$ above which the ensemble outperforms the single large model of size $N$ as at least $M/K$.   This bound on appears to be tight in the overparameterized regime where $P \ll N$ (fig. S2.C, F).  We state this result explicitly as the new "Corrolary 1" in the updated version of the manuscript.
>
> > One might hope to apply the result from section 9.1 of [1] or from [4] to prove our Theorem 2 by formally regarding the projection of $M$ random features as the full data, with correspondingly shifted ground-truth weights
> $\tilde{\bar{\mathbf{w}}} = \text{argmin}\_\{\mathbf{w}\} \left[ \mathbb{E}\_\{\mathbf{x} \sim \mu\_\{\mathbf{x}\}\} (f_*(\mathbf{x}) - \mathbf{w}^\top \mathbf{Z} \mathbf{\theta}(\mathbf{x}))^2 \right]$
> and shifted label noise
> $\tilde{\sigma}\_\epsilon^2 = \sigma\_\epsilon^2 + \mathbb{E}\_\{\mathbf{x} \sim \mu\_\{\mathbf{x}\}\} \left[ \left( f\_\ast(\mathbf{x}) - \tilde{\bar{\mathbf{w}}}^\top \mathbf{Z} \mathbf{\theta}(\mathbf{x}) \right)^2 \right]$
> With $NK = M$, the ensemble of $K$ RFRR models of size $N$  could formally be regarded regarded as sampling mutually exclusive subsets of $N$ features from the pool of $M$. However, this requires the regressors to sample non-overlapping subsets of the features, which is not accomodated by the random bernoulli masks in [9, 3] or the uncorrelated sketching matrices in [4].  For these reasons, we believe that our Theorem 2 is not a trivial result.
>
> [1] Srivastava, N., Hinton, G., Krizhevsky, A., Sutskever, I., & Salakhutdinov, R. (2014). Dropout: a simple way to prevent neural networks from overfitting. The journal of machine learning research, 15(1), 1929-1958.
>
> [2] Zhang, J., & Bottou, L. (2023, July). Learning useful representations for shifting tasks and distributions. In International Conference on Machine Learning (pp. 40830-40850). PMLR.
>
> [3] Daniel LeJeune, Hamid Javadi, Richard Baraniuk Proceedings of the Twenty Third International Conference on Artificial Intelligence and Statistics, PMLR 108:3525-3535, 2020.
>
> [4] Patil, Pratik, and Daniel LeJeune. "Asymptotically free sketched ridge ensembles: Risks, cross-validation, and tuning." arXiv preprint arXiv:2310.04357 (2024). Available at: https://arxiv.org/abs/2310.04357.

---

> ### Author Response · Authors · 2024-11-21
> **Response to Reviewer Comments (Part 2)**
>
> The cifar experiments in this work does not link to the theory, i.e. without neither regression nor random features.
> > **Response**: Thank you for this comment, which prompts a clarification.  First, we note that because RFRR is still a commonly used algorithm, our theoretical contributions to understanding RFRR may be of independent interest.  Our theory of RFRR ensembles does link directly to the Binarized CIFAR10 and MNIST RFRR experiments, which are overlaid with theoretical loss curves in figures 1, 2, and 4.
>
> >We do not claim that our theoretical results directly explain our empirical findings in deep neural networks (fig. 5).  Rather, we view the RFRR model as a toy model from which we might gain intuition that guides our numerical experiments.  This perspective has guided many papers (for one example, see [5]).  In this case, we find that RFRR ensembles obey the "no free lunch from ensembles" principle when the hyperparameter $\lambda$ is tuned to its optimal value.  In section 6, we find that deep ensembles trained with $\mu P$ parameterization also obey the ``no free lunch from ensembles'' principle, when network hyperparameters are tuned to their optimal values.  The difference is that the hyperparameters which must be tuned in deep networks are both weight decay and richness $\gamma$. Please see our updated section 6, which makes the relationship between theory and deep feature-learning ensembles clearer than in our original submission.  We agree that a theory of feature-learning ensembles is an important objective for future work, and have added a note on this to our discussion section.
>
> Questions:
> The CNN experiment on CIFAR10 contains two CNN layers and one linear layer. That is already too small for CIFAR10 tasks. Reducing the size of this small network would further hurt the performance. Does the same phenomena in this paper hold on larger networks? [2] shows the ensemble multiple small networks outperforms a large network.
>
> > **Response**: Thank you for this comment and for bringing the results of [2] to our attention.  We are currently working on repeating the experiment in fig. 4A using a larger ResNet architecture that performs better on the CIFAR10 classification task.  We will hopefully be able to share these before the end of the discussion period.  As for the results of [2], they do not appear to be inconsistent with our claims.  This is because [2] does not use the $\mu P$ parameterization.  Many of their results might be explained by the fact that in standard parameterization, a single wider network will be closer to the lazy learning regime than an ensemble of smaller networks.  It is possible (and we suspect) that some of the benefits of ensembling that they report would be reversed if they used $\mu P$ parameterization and optimized network performance over the richness parameter $\gamma$ as well as weight decay in each experiment.  We have updated section 6 to elaborate on the importance of disentangling network size and learning dynamics to make a proper comparison between networks of varying width.
>
> [2] Zhang, J., & Bottou, L. (2023, July). Learning useful representations for shifting tasks and distributions. In International Conference on Machine Learning (pp. 40830-40850). PMLR.
>
> [5] Preetum Nakkiran, Prayaag Venkat, Sham Kakade, and Tengyu Ma. Optimal regularization can mitigate double descent. arXiv preprint arXiv:2003.01897, 2020.

---

> ### Comment · Reviewer_6nYe · 2024-11-26
>
> Thank for authors responses. After reading the responses and other reviews comments, I find some key concerns are not well addressed.  In summary:
>
> 1) The theorical result is trival (close to the first two weakness pointed by reviewer **sshy**). The main theorical result in this work is still *"one right regression on random features size M is always the optimal under the optimal L2 regularization, compared with the ensemble of K ridge regressions on random features size M/K."*. This is trival because there is a L2 regularization to trade-off the model complexity.
>
> 2) **Text, Theory and Experiments are not consistent**, as author replied *"We do not claim that our theoretical results directly explain our empirical findings in deep neural networks (fig. 5)."*
>
> 3) The CNN experiment on Cifar is problematic, because the network of each ensemble member is highly constrained (too small).
>
> Overall, I couldn't recommend for acception.

---

> > ### Author Response · Authors · 2024-11-27
> >
> > Thank you for considering our rebuttal.  First, to your concern about our CNN experiments, we remind you that we have repeated these experiments in ResNet18 ensembles with about 30 million parameters, finding similar results to the smaller CNN ensembles.  Please see our revised Figure 5, panel b.  We will also try one more time to explain why we think our results are nontrivial and interesting.
> >
> > We have provided a sound argument for why our main result is not trivial in our last rebuttal.  In particular, we have shown that if any of the assumptions of our Theorem 2 are broken, the result no longer holds:
> > - If there is not an optimal L2 regularization, then the ensemble of $K$ models of size $N$ might outperform a single model of size $M=NK$ (see non-monotonic curves in figure 2A).
> > - If $K>M/N$, even by a tiny amount, it is possible for an ensemble of $K$ models of size $N$ to outperform a single model of size $M$ even at optimal ridge (see the newly addes figure S2, panels C, F).  The theorem would certainly not hold for an exponential number of small regressors.
> >
> > We believe that this makes our result non-trivial, in that it could not possibly be made any stronger -- any relaxation of the assumptions would result in the theorem breaking.  Whether or not this result is surprising is a subjective question.  We see scientific merit in proving this statement, as intuitions often turn out to be wrong.
> >
> > Finally, our theory **does** directly explain our experiments in ReLU random-feature ensembles (figures 1, 2, 3, 4), as well as our experiments in deep networks in the lazy regime (small $\gamma_0$ in figure 5).  The only part of our experiments not covered by the theory are the rich-regime learning curves for deep network ensembles.  Including these, we believe, strengthens our paper by showing how feature-learning effects breaking the assumptions of our theorems may complicate the behavior of ensembles.  Our empirical observations in feature-learning ensembles are much more compelling against the backdrop of our analytical results for lazy networks than they would be on their own.  The theory allows us to pinpoint any violations of the "more is better" or "no free lunch from ensembles" behaviors in deep ensembles as a pathology of feature learning.

---

> > > ### Author Response · Authors · 2024-12-02
> > >
> > > To clarify our comment stating that if $K>M/N$, even by a tiny amount, it is possible for an ensemble of $K$ models of size $N$ to outperform a single model of size $M$ even at optimal ridge, we will add to the final version of our paper an asymptotic expansion of the risk $E_g^K$ at large $N$ and small ridge $\lambda$:
> > >
> > > $$ E_g^K = -\frac{P {\kappa_2^*}^2 \operatorname{tf}_1'(\kappa_2^*)}{P - \operatorname{Df}_2(\kappa_2^*)} + \lambda  F(\kappa_2^*, P) + \frac{P \kappa_2^* \operatorname{tf}_1(\kappa_2^*)}{KN} + O (\lambda^2, \lambda/N, 1/N^2)$$
> > >
> > > where $\operatorname{Df}_1(\kappa_2^*) = P$. Here, you can see that at leading order, the error of an overparameterized ensemble depends on ensemble size $K$ and model size $N$ only through the total number of features $KN$.  It follows that when $KN>M$, even by a tiny amount, an ensemble of $K$ models of size  $N$ can outperform a single larger model of size $M$  provided $P \ll N$ and optimal or near-optimal performance can be achieved with a small ridge $\lambda$.
> > >
> > > [1] Ben Adlam and Jeffrey Pennington. Understanding double descent requires a fine-grained bias-variance decomposition, 2020. URL https://arxiv.org/abs/2011.03321.

---

### Official Review · Reviewer_SSHY · 2024-11-04

**Soundness:** 3
**Presentation:** 2
**Contribution:** 1
**Rating:** 3
**Confidence:** 3

**Summary:**

This paper investigates the trade-off between employing a single large model versus an ensemble of smaller models, focusing on random feature kernel ridge regression (RF-KRR). In particular, this work studies ensembles of size $K$ with $N$ random features per ensemble member with a fixed total parameter count $M=N\cdot K$. In this setting, the paper proves rigorously and shows empirically that optimal performance is achieved by $K=1$ for an optimally tuned ridge parameter while increasing $K$ degrades the optimal test risk. Additionally, this result – referred to as “_no free lunch from random feature ensembles_” – is shown for CNNs and transformers in experiments on image classification (CIFAR10) and language modelling (C4) in the regimes of lazy training and feature learning. Furthermore, scaling laws are derived, and conditions for achieving near-optimal scaling laws for these ensembles are identified.

**Strengths:**

The main contribution of the paper is to add the ensemble perspective to established results on random feature kernel ridge regressions. It contributes theoretical insights and proofs for two theorems on “_more is better for RF Ensembles_” and “_No Free Lunch From Random Feature Ensembles_” (pages 4-5)  as well as scaling laws and empirical validation of adequate quality. Generally, the paper has a clear thread, with some minor adjustments required to the presentation (listed below in “weaknesses”).

**Weaknesses:**

Even though extending the analysis of RF-KRR to ensembles is new, the presented results align with the expectations from prior work on single models. In my opinion, this limits the novelty and significance of the contribution, which I elaborate on in more detail below.

__W1.__ _Novelty of Theorem 1 “More is better for RF Ensembles” and results in section 3_: For “non-ensemble” RF-KRR, the referenced papers by [Simon _et al_. (2023)](https://openreview.net/pdf?id=OdpIjS0vkO) and [Bordelon _et al_. (2024a)](https://arxiv.org/pdf/2402.01092) have established that decreasing the number of random features $N$ leads to a degraded optimal test risk. Could the authors elaborate on why examining ensembles adds new perspectives beyond what is already known for single models? I think a strong justification would be necessary. In particular, I view the statement of Theorem 1 and the results presented in Figure 1 of this submission as a natural consequence of (and essentially the same as) [Theorem 1 “More is better for RF regression” and Figure 1 in in Simon _et al_. (2023)](https://openreview.net/pdf?id=OdpIjS0vkO#page=6); see also point W2.1.

__W2.__ _Significance of Theorem 2 “No Free Lunch From Random Feature Ensembles” and results in section 4_: \
__W2.1.__ A main premise of the work is to have a fixed total parameter count $M$ and compare ensembles of $K$ members and $N$ random features such that $M=N\cdot K$ always holds. This is presented as a “pragmatic” approach to limit computational overhead associated with ensemble methods (lines 34-37). However, this leads to increasingly “weaker learners” when the ensemble size $K$ is increased. For this reason, it is less surprising and in line with the conclusions of [Simon _et al_. (2023)](https://openreview.net/pdf?id=OdpIjS0vkO) and [Bordelon _et al_. (2024a)](https://arxiv.org/pdf/2402.01092) that decreasing the number of random features $N$ (by increasing ensemble size $K$) leads to a degraded optimal test risk (at fixed total parameter count $M$). From this perspective, it is expected that $K=1$ leads to the optimal result. This limits the novelty and significance of theorem 2 on “No Free Lunch From Random Feature Ensembles” where specifically the number of random features are chosen as $N’=M/K’$ and $N=M/K$ with $K’<K$ leading to $N’>N$.  As this is a central result of the paper, it could strengthen the contribution to elaborate more on why ensembles provide new insights beyond what can be directly inferred from the existing work on single models. In my opinion, a more substantial justification would be required. \
__W2.2.__ In addition to the previous point, I believe that the interplay between the number of random features $N$, the size of the ensemble $K$ and the total parameter count $M$ make it difficult to compare the information provided in Figures 1 and 2. For instance, in Figure 2A the relationship between test error and ensemble size $K$ for different sample sizes $P$ at fixed total parameter count $M$ is considered, but with increasing $K$ the number of random features $N$ decreases. It appears that a more careful discussion would be required, which is intimately connected to the premise above.

Minor remarks:
- “RF KRR” and “RF-KRR” are both used in the manuscript and the authors might want to make the usage consistent.
- Line 101: Missing full stop “.” at the end of the sentence.
- Line 106: The computation of $g(v_n,x)$ is mentioned, but $g$ is not defined in the main text. From Appendix A, line 690, it seems that it is another notation for $\psi^k (x)$, but its use in the main text is not clear to me.
- Line 118-119: Missing citation.
- Line 142: Wrong formatting of “f_{*} (x)”.
- Line 176-177: Double parenthesis in citation.
- Line 187: I would not recommend writing “eq’s”.
- Line 248-249 and Figure 1: The “filled” lines are quite likely supposed to be dashed lines.

In summary, I view the points raised in W1 and W2 as the main challenges in the current version of the submission and believe a major revision of the paper is required. However, I invite the authors to address my objections and clarify potential misunderstandings.

**Questions:**

My questions revolve around the core assumptions of the paper as outlined in W1 and W2 above. In particular, considering classical literature on RF-KRR like [Rahimi and Recht, “Weighted Sums of Random Kitchen Sinks: Replacing minimization with randomization in learning” (2008)](https://people.eecs.berkeley.edu/~brecht/papers/08.rah.rec.nips.pdf) and the referenced works of [Simon _et al_. (2023)](https://openreview.net/pdf?id=OdpIjS0vkO) and [Bordelon _et al_. (2024a)](https://arxiv.org/pdf/2402.01092), my questions are as follows:

Q1. Is there any evidence that an ensemble of random features could provide a different result than is stated in [Theorem 1 of Simon _et al_. (2023)](https://openreview.net/pdf?id=OdpIjS0vkO#page=6) for RF regression?

Q2. In connection to Theorem 2 in the submission and in the context of RF-KRR, is there any reason to assume that a larger ensemble of weak learners can outperform a small ensemble of stronger learners?

---

> ### Author Response · Authors · 2024-11-21
> **Response to reviewer comments.**
>
> Weaknesses:
>
> W1. Novelty of Theorem 1 “More is better for RF Ensembles” ... see also point W2.1.
>
> > **Response**: Thank you for this comment.  While we agree that this result is in line with results from Simon et al. (2023) and Bordelon et al. (2024a) for single models, we believe that it might be surprising in the context of ensemble learning, where predictive variance has historically been viewed as beneficial.  In random forests, for example, subsampling of data dimensions leads to improved performance even though it reduces the sizes of individual decision trees.  In RFRR ensembles, each ensemble member is distinguished by the particular realization of its random features (i.e. the independently drawn $\mathbf{v}\_n\^k \sim \mu\_\{\mathbf{v}\}$).  As $N \to \infty$, the function learned by each estimator will converge to the same limiting kernel predictor, destroying the diversity of the ensemble.  One might therefore expect *reducing* the size $N$ of each ensemble member to improve ensemble performance by increasing the diversity of the ensemble's predictors. Other empirical findings have shown that deep feature-learning ensembles do not always conform to the ``wider is better'' intuition observed for single models (see fig. 8 from  [1]).  We have updated the text of section 3.3 to discuss these points.
>
> W2. Significance ... section 4:
> W2.1. A main... would be required.
>
> W2.2. In addition to ...  premise above.
>
> > **Response** Thank you for your comments, which prompt an important clarification of the difference between figures 1 and 2.  As you have correctly understood, in figure 1 we compare ensembles across widths $N$ while keeping ensemble size $K$ fixed.  We have made a slight alteration to our numerics to make this clearer: the x-axes of figure 1B now correspond to the width $N$ of the ensemble members.  In this case, the total parameter count $M = KN$ grows with $N$.  In line with our expectations from Simon et al. (2023) and Bordelon et al. (2024a), we see that the optmial risk decreases with $N$.  However, we argue that in the case of ensembles of predictors, this does not follow directly from their results for single models, and might be surprising given the common practice of feature-bagging (see our response to W1).  In figure 2 we compare the ridge-optimized performance of RFRR models with fixed total parameter count $M=KN$ as the ensembe size $K$ is varied.  You have correctly understood then that as $K$ increases, the size $N$ of each ensemble member decreases ($N = M/K$).  We disagree, however, that we should necessarily expect error to increase as a result of this decrease in $N$ because $K$ is also increasing, which is beneficial to the error. Put plainly, the decrease in $N$ and increase in $K$ are competing effects.  If variance is the main contribution to the error, one might expect to see a benefit to reducing model size if that means you can average over a larger number of predictors.  Theorem 2 guarantees that this is never the case in RFRR, provided that the total parameter count is fixed and ridge is optimized.  An ensemble of smaller models might do better than a single larger model, however, when the total parameter count of the ensemble is larger than the number of parameters in the single larger model.  We have formalized this fact in the corollary added to section 4 and the surrounding discussion.  Please also see the newly added figure S2.
>
> Minor remarks:
>
> “RF KRR” ... dashed lines.
>
> > **Response**: Thank you for pointing out these errors, which have been corrected in the new version of the manuscript.  We have replaced "eq's" with "equations."  The text of section 2 has been updated to clarify the relationship between $\mathbf{\psi}\(\mathbf{x}\)$ and $g\(\mathbf{v}\_n, \mathbf{x}\)$.
>
> Questions:
> Q1. Is there any evidence ... RF regression?
>
> > **Response**: We refer to our response to W.1 above.
>
> Q2. In connection to Theorem 2 i... stronger learners?
>
> > **Response**: We refer to our response to W.2 above.  In particular, we reiterate that a larger ensemble of weak learners *can* outperform a small ensemble of stronger learners.  This is only possible, however, when the total parameter count of the larger ensemble of weak learners is greater than the total parameter count of the smaller ensemble of strong learners.
>
>
> [1] Nikhil Vyas, Alexander Atanasov, Blake Bordelon, Depen Morwani, Sabarish Sainathan, and Cengiz Pehlevan. Feature-learning networks are consistent across widths at realistic scales, 2023.
> URL https://arxiv.org/abs/2305.18411.
>
> [2] Leonardo Defilippis, Bruno Loureiro, and Theodor Misiakiewicz. Dimension-free deterministic equivalents and scaling laws for random feature regression, 2024. URL https://arxiv.org/abs/2405.15699.

---

> > ### Comment · Reviewer_SSHY · 2024-11-24
> >
> > I appreciate the authors’ effort to address my objections and similar objections raised by other reviewers (in particular, __U2xD__ and __6nYe__) and believe that the additional experiments strengthen parts of the contributions in this submission. However, my initial reservations about some of the main aspects at the beginning of the paper persist, partly because I feel that my previous objections were not directly addressed:
> >
> > __Regarding the response to W1 (Novelty of Theorem 1) and question Q1__: In my opinion, some considerations and intuition from the feature-learning setting are mixed with the random feature setting. Similar points were raised by reviewers __U2xD__ and __6nYe__ to some extent. In particular, the objections and questions of weakness W1 and Q1 were explicitly stated in the context of random feature kernel ridge regression. However, the provided answer relies on insights from feature-learning models, like the decreased performance of “wider” deep feature-learning ensembles. While I agree that the response is sensible when talking about models that learn features, I disagree that this naturally extends to random feature models. In this regard, I struggle to see the justification of the hypothesis that  “_[o]ne might therefore expect __reducing__ the size of each ensemble member to improve ensemble performance by increasing the diversity of the ensemble's predictors_" when features are extracted at random for each ensemble member and the KRR is __regularised with the optimal ridge__. In this explicit context of RFRR, could the authors provide some examples/evidence that justifies this hypothesis and shows a gap to the results by Simon _et al._ (2023)?
> >
> > __Regarding the response to W2 (Significance of Theorem 2) and question Q2__: My  (broad) objection here aligns with the more detailed comment provided by reviewer __6nYe__. Similar to what I wrote above, I believe the implicit hypothesis leading to theorem 2 requires more evident justification. The authors state that they “_[…] disagree, however, that we should necessarily expect error to increase as a result of this decrease in $N$ because $K$ is also increasing, which is beneficial to the error._” In the explicit context of RFRR regularised with optimal ridges and the premise of the submission ($N\times K = M$ with fixed $M$), I am curious to understand the motivating results/evidence that leads to the disagreement and justifies the underlying hypothesis addressed by theorem 2? In essence, this is what I was trying to ask with question Q2. The provided answer where one deviates from the “fixed-parameter-count” premise was not contested.
> >
> > Although I agree that there is some extension of the established results and theorems, I still consider this part of the paper (presented as a main contribution) of limited novelty and significance for the above reasons.

---

> > > ### Author Response · Authors · 2024-11-25
> > >
> > > Regarding the response to W1 (Novelty of Theorem 1) and question Q1: In my opinion, some considerations... shows a gap to the results by Simon et al. (2023)?
> > >
> > > > **Response** Thank you for considering our responses, we appreciate the opportunity to clarify the motivation for and significance of our results.  As you have correctly understood, our Theorem 1 reduces to the result of Simon et. al. in the special case $K=1$.  The "bigger is better" result for an ensemble does not, however, follow directly from the $K=1$ case.  Proving "bigger is better" for RFRR *ensembles* at optimal ridge is a novel contribution. Whether or not this result is surprising is a subjective question -- either way, there is scientific merit in proving it as intuitions often do not hold true when put to the test.
> > >
> > > > Your review asked us to motivate Theorem 1 as a surprising result. We have cited feature-learning ensembles as a class of models in which "wider is better" for $K=1$ does not extend to "wider is better" for ensembles (see figure 8 from [1]).  Of course, this does not naturally extend to RFRR ensembles at optimal ridge -- if it did, theorem 1 would not be true!  Nevertheless, the example of feature-learning ensembles demonstrates that even if "wider is better" holds for a single model, it doesn't necessarily hold for an ensemble of the many models of the same type.  This furthers our argument that Theorem 1 is a novel result, which does not follow directly from the results of Simon et. al. (2023).
> > >
> > > > We also want to emphasize that theorem 1 is not our main result.  We agree that this result may not surprise those familiar with the results of Simon et. al. (2023), even if it was not stated or proven there.  Theorem 2 and Corollary 1, however, are much more compelling results, and the focus of all figures following figure 1.  We have included theorem 1 and figure 1 because it is necessary for completeness, because it is necessary to corollary 1, and because Simon et. al. (2023) did not address the distinct case of ensembles.
> > >
> > > Regarding the response to W2 (Significance of Theorem 2) and question Q2: My (broad) objection here aligns with ... answer where one deviates from the “fixed-parameter-count” premise was not contested.
> > >
> > > > **Response** Plenty of papers have claimed that performance gains can be obtained by dividing the number of parameters in one neural network into an ensemble of smaller networks.  Our hypothesis is that ensembling is not as useful as simply scaling up the size of a single predictor when network hyperparameters are tuned to their optimal values.  In RFRR, there is only one hyperparameter to be tuned -- the ridge. In line with our hypothesis, we find that when this hyperparameter is tuned to its optimal value, the optimal strategy is to combine all parameters into a single large model.  In feature-learning networks, there are more hyperparameters to be tuned, but again in line with our hypothesis we find that at optimal weight decay and richness the "no free lunch from ensembles" principle holds.  We understand that our theory does not directly explain the behavior of feature-learning ensembles, and have updated our manuscript to make this clear.  However, we believe that these results should be presented side by side because of their similar flavor, and because of the correspondence between random-feature models and deep ensembles in the lazy training regime.
> > >
> > > > Furthermore, we believe Theorem 2 to be a compelling result because it represents a fundamentally new type of monotonicity result that applies to the allocation of a fixed set of resources rather than the amount of available resources for a learning problem.  The "more is better" results from Simon et. al. and our Theorem 1 compare networks or ensembles with a different total number of random features (or samples).  It's not a surprise that when optimal regularization is used to mitigate over-fitting effects, having more resources is beneficial to generalization.  Our theorem 2 instead informs the way that a fixed budget on resources is *used*.  Saying that a fixed number of random features is better when used together in a single model than divided into an ensemble is a novel type of statement relative to saying that more random features *total* is better than fewer random features *total*.  In fact, this type of comparison is only possible in the context of ensembles of predictors.  We hope this clarifies the novelty of our contribution, and justifies our focus on Theorem 2 as the main result of our paper.

---

> > > > ### Comment · Reviewer_SSHY · 2024-11-29
> > > >
> > > > Thank you for the responses, but this discussion goes in circles without addressing the main point. From the narrative of the paper and the authors’ responses, it becomes clear that much of the motivation is borrowed from insights of feature learning models. However, the paper positions the main theorems and main contributions in the context of random feature models. I perceive this as an inconsistency that pertains to the empirical evaluation, too. Arguing that because some results hold for the feature learning setting, a similar hypothesis should hold for the random feature setting is a relatively weak justification for the hypotheses leading to Theorem 1 and 2, in my opinion. This is a main reservation regarding the submission which has not been addressed so far. Based on my overview of the literature on random feature models, the submission does not address a significant gap in the existing literature. I acknowledge the extension to ensembles but view the rationale behind why the ensemble approach should lead to significantly different and novel results than those provided in the highlighted literature as insufficiently justified.

---

> > > > > ### Author Response · Authors · 2024-12-02
> > > > >
> > > > > Thank you for clarifying your objections.  We would be happy to shift the narrative focus of our paper to center random-feature models as well as older ensemble-based classification algorithms (i.e. random forests) as the primary motivations for our theoretical results.  Importantly, feature-bagging (where each ensemble member is trained on a subset of available features) is a well-established practice for ensemble learning.  We believe it will surprise a great many readers that this is provably detrimental to RFRR, especially in the context of Theorem 2 where subsampling (reducing $N$) has the added benefit of increasing ensemble size $K$.  We cannot, however, provide a sound argument for why these results should be "surprising" *for RFRR models at optimal ridge* because, at the end of the day, they are true in this context.
> > > > >
> > > > > Even though our theoretical results do not explain the behavior of feature-learning ensembles, we believe that they are an important step toward understanding deep ensembles because of the correspondence between RFRR and deep networks in the lazy regime.  In parcitular, our results allow us to pinpoint any violations of the "more is better" or "no free lunch from ensembles" principles in deep networks at optimal weight decay as a consequence of feature learning specifically.  We will update the main text to clarify this motivation.  Furthermore, there is a long history of research on random-feature models in analogy to deep networks.  Another example of this fruitful correspondence is work on fine-grained bias-variance decompositions [1].  Could your criticism that we are asking questions motivated by observations made in deep networks in the context of RFRR not just as easily be applied to Simon et. al. (2023)?
> > > > >
> > > > > We have also shown that if any of the assumptions of our Theorem 2 are broken, the result no longer holds:
> > > > > - If there is not an optimal L2 regularization, then the ensemble of $K$ models of size $N$ might outperform a single model of size $M=NK$ (see non-monotonic curves in figure 2A).
> > > > > - If $K>M/N$, even by a tiny amount, it is possible for an ensemble of $K$ models of size $N$ to outperform a single model of size $M$ even at optimal ridge (see the newly addes figure S2, panels C, F).  We are not aware of any previous studies which have pointed this out.  To drive this point home, we will add to the final version the asymptotic expansion of the risk $E_g^K$ at large $N$ and small ridge $\lambda$: $ E_g^K = -\frac{P {\kappa_2^*}^2 \operatorname{tf}_1'(\kappa_2^*)}{P - \operatorname{Df}_2(\kappa_2^*)} + \lambda  F(\kappa_2^*, P) + \frac{P \kappa_2^* \operatorname{tf}_1(\kappa_2^*)}{KN} + O (\lambda^2, \lambda/N, 1/N^2)$, where $\operatorname{Df}_1(\kappa_2^*) = P$. Here, you can see that at leading order, the error of an overparameterized ensemble depends on ensemble size $K$ and model size $N$ only through the total number of features $KN$.
> > > > >
> > > > > We believe that this makes our result non-trivial, in that it could not possibly be made any stronger -- any relaxation of the assumptions would result in the theorem breaking.
> > > > >
> > > > > [1] Ben Adlam and Jeffrey Pennington. Understanding double descent requires a fine-grained bias-variance decomposition, 2020. URL https://arxiv.org/abs/2011.03321.

---

> > > > > > ### Comment · Reviewer_SSHY · 2024-12-03
> > > > > >
> > > > > > Thank you for the responses. Regarding the authors’ question whether my criticism could not as easily be applied to Simon et al. (2023): It is not contested that studying random feature models can provide valuable insights into feature-learning models (and vice versa), nor that results in one area might motivate research questions in the other (and vice versa). In that regard and in context of overparameterised models, Simon et al. (2023) provide relevant and rigorous contributions for random feature models. However, as these results are already established, my main objection in the initial review and follow-up comments is how much the ensemble perspective adds to the existing literature, specifically Simon et al. (2023), and whether a meaningful gap was addressed. Thus, my justification request in previous comments was targeted at the explicit context of the theorems, specifically RF-KRR (and not to provide “counterexamples” for theorem 2). It is not immediately apparent to me how the suggested (future) adjustments in motivation from random forests fits in the general context of the paper (deep learning and RF-KRR).
> > > > > >
> > > > > > For these reasons, I stay with my initial assessment. In my opinion, a major revision of the submission addressing the highlighted aspects would sharpen the contributions and provide more conclusive insights. However, if the other reviewers feel strongly about the contributions of this paper, I will not object.

---

### Official Review · Reviewer_U2xD · 2024-11-07

**Soundness:** 3
**Presentation:** 4
**Contribution:** 3
**Rating:** 6
**Confidence:** 2

**Summary:**

- This paper studies the tradeoff between ensemble size and total feature/parameter count for random feature ensembles
- In particular, they derive the following theorems (paraphrased informally):
- Theorem 1 - larger ensembles -> better performance (this contribution is minor)
- Theorem 2 - Given a fixed feature count, it is better to have a single large model than multiple smaller ones (this is the main contribution of the paper)
- Both theorems are verified experimentally binarized Cifar-10 datasets
- The paper then uses Thm 2 to derive scaling laws, assuming a standard model of the decay of kernel's eigenspectrum. They identify differing behaviour for "simple" and "difficult" datasets, and verify this experimentally
- Finally, the paper studies the performance of non-fixed feature classifiers using ensembles of CNNs trained on CIFAR-10. They control the feature learning by controlling the richness parameter

**Strengths:**

- Presentation is clear and easy to follow
- Claims are verified experimentally and there is good agreement with the author's claims
- Section 5 (the section on scaling laws) I find particularly interesting as it shows how the fairly uninspiring theorems in the first two sections can be used to derive more non-trivial behaviours, such as the distinction between "easy" and "difficult" dataset scaling laws
- The use of the richness parameter to control the adherence to the assumptions in the theorems in section 6 in quite interesting, but I am unsure how applicable this work is necessarily to modern models (see weaknesses)

**Weaknesses:**

- I think the authors should be very clear that the theorems do not directly "explain" the empirical results in section 6, except in the case of very small richness parameters, as their theorems do not cover feature learning. I think it would also be useful for the authors to discuss this limitation in the weaknesses section in concluding discussion

**Questions:**

- Is it possible to plot the validation loss in figure 5b as opposed to the test loss?
- Could the authors discuss how for certain values for $\gamma$ in figure 5a, it seems the increasing the ensemble count improves performance, namely for larger values of $\gamma$? Is it possible that behaviour for models outside of the lazy regime do not follows the same "no free lunch" theorem?

---

> ### Author Response · Authors · 2024-11-21
> **Response to reviewer comments**
>
> - Weaknesses:
>     - I think the authors should be very clear that the theorems do not directly "explain" the empirical results in section 6, except in the case of very small richness parameters, as their theorems do not cover feature learning. I think it would also be useful for the authors to discuss this limitation in the weaknesses section in concluding discussion.
>     > **Response** We thank the reviewer for this suggestion, which we agree with. We have made significant updates to clarify that our theory does not cover feature learning in both section 6 and in the discussion section.
>
> - Questions:
>     - Is it possible to plot the validation loss in figure 5b as opposed to the test loss?
>     > **Response**: Thank you for this suggestion. We clarify that Figure 5b shows the training loss for the C4 language modeling task as a function of training steps in the online setting.  In this setting, there is no distinction between train and test loss as each batch consists of previously unseen data.  Calculating a validation loss on a larger held-out test set might reduce noise in this figure, but because the curves are already visually smooth, we do not believe this will make a significant difference in the resulting plot.  We will add text clarifying the meaning of the loss in the online setting.
>     - Could the authors discuss how for certain values for $\gamma$ in figure 5a, it seems the increasing the ensemble count improves performance, namely for larger values of $\gamma$?  Is it possible that behaviour for models outside of the lazy regime do not follows the same "no free lunch" theorem?
>     > **Response**: Thank you for pointing this out.  After checking our experiments, we agree that models outside the lazy regime do not necessarily follow the "no free lunch" principle.  However, we do find empirically that monotonicity is restored when both the weight decay and the richness parameter are jointly tuned to their optimal values.  Please see the updated figure 5a and text of section 6.

---

> > ### Comment · Reviewer_U2xD · 2024-12-02
> >
> > I have followed the discussion that the author and reviewers have been having. I still maintain my position that this work is a valuable contribution, despite its limitations wrt feature learning models. I believe that explaining feature learning models would be outside the scope of this work. I will be maintaining my score at 6.

---

### Official Review · Reviewer_knrL · 2024-11-12

**Soundness:** 3
**Presentation:** 3
**Contribution:** 3
**Rating:** 8
**Confidence:** 2

**Summary:**

The present study aims to investigate the utility of training multilple shallower networks vs that of a larger network.  Theory shows that only 1 large network is appropriate and experiments validate this approach.

**Strengths:**

The paper gives extensive theory and experimental proof that the theory works.

Wide variety of experiments.

Scaling laws are given for this class of models, which is always useful.

**Weaknesses:**

I'd like to see experiments be ran on imagenet validating these results there.

No results on conventional nerual network architectures.

**Questions:**

Would it be possible to train random feature ResNet18s or VGGs?

---

> ### Author Response · Authors · 2024-11-21
> **Response to reviewer comments**
>
> Thank you for your review!  To your question of using conventional neural network architectures, we have also validated these results for CIFAR10 calssification with ResNet18 ensembles (See Fig. 5.B in updated manuscript).

---

### Official Review · Reviewer_mDQV · 2024-11-12

**Soundness:** 3
**Presentation:** 3
**Contribution:** 3
**Rating:** 8
**Confidence:** 2

**Summary:**

The paper covers several theoretical and empirical analyses related to whether a parameter budget should be allocated to a single large classifier or an ensemble of smaller classifiers.
The authors first analyse ridge regression using random-features where the ensemble prediction is the mean response of the member predictors, and the test risk is the mean squared error.  They prove that the expected test risk decreases when any of: data size, ensemble feature size, or ensemble size increases when the ridge parameter is chosen optimally for the given setting (with mild assumptions on the task eigenstructure) and show empirically the monotonicity in test error for a binarized CIFAR10 task using RELU random features. The authors then prove a “no free lunch” theorem for random feature ensembles showing that, for a fixed total parameter count, lower test risk is achieved with fewer ensemble members (with mild assumptions on the task eigen structure) and empirically show the monotonicity of test error with ensemble size fitting RELU random feature models to the binarized CIFAR10 task.
Next the authors derive scaling laws for random feature ensembles in the width-bottlenecked regime (the number of features per ensemble member is much smaller than the data size). They show on synthetic data that for r (a parameter controlling the relative rate of decay of power in modes relative to the rate of eigenspectrum decay) greater than 0.5 (an “easy task”) then there is a growth exponent l* above which scaling laws are near optimal. Therefore ensembles with K>1 can be near optimal as long as the number of parameters per ensemble member grows quickly enough.
Finally, the authors initialize CNNs in the maximal update parameterization and, using CIFAR10, empirically show for a fixed parameter budget, and task richness, that test accuracy typically decreases monotonically with increased ensemble size. They also show empirically that training loss increases monotonically with increased ensemble size when fitting transformers to C4 data.

**Strengths:**

I think the paper is in general clear and well-written, the research question is useful and the paper is of interest to ICLR.
I have not checked the proofs, but the results looked sensible and were broadly consistent with the empirical results.

**Weaknesses:**

In figure 5a, one of the highest performing richness parameters (although not the admittedly highest) does not show monotonic decrease with ensemble size. Given this does not fit with your other analysis, it would be useful to comment on it in the paper.



minor points:

Both CIFAR10/MNIST and a CIFAR10/MNIST-derived binary classification task are included in the paper. What is the papers convention for differentiating between the binarized version and the original version? In places the paper appears to call the binary classification task “a CIFAR classification task” or “Binarized CIFAR10” but in others it then refers to this binary task just as CIFAR (I think this happens for example in line 370). Could the binary tasks have a name (for which CIFAR10 is a prefix perhaps) and then be referred to by that name to improve readability?

The same notation of learning rate and task eigen structure, and for richness and the data-size-scaled degree of freedom parameters. Giving separate notation would be preferable.

118 – “(citations)” should instead be the actual citation

228 – Figure 1: Are the non-red lines supposed to be dashed in this plot? otherwise does the legend contain a dashed line? Since none appear in the actual plot.

340 – Figure 3 – could you draw the line for l* (= 1/(1+ alpha*(2*r – 1))) in red on the plots for figure 3 A?

397 – “a” is singular, but “tasks” is plural, should remove “a”?

452 – “that at the” -> “that the”

**Questions:**

In Figure 2c, ensembles (K>1) appear to be relatively robust to ridge parameter, where as K=1 is highly sensitive to it. Could this correspond to it potentially being more practically expedient to train small ensembles when optimally tuning the regularising parameter is expensive?

---

> ### Author Response · Authors · 2024-11-21
> **Response to reviewer comments**
>
> Weaknesses:
> In figure 5a, one of the highest performing richness parameters (although not the admittedly highest) does not show monotonic decrease with ensemble size. Given this does not fit with your other analysis, it would be useful to comment on it in the paper.
>
> > **Response**: Thank you for this comment, which we agree with.  We have updated our discussion of the results of our deep ensemble simulations to discuss these exceptions to the "no free lunch" principle.  We observe that monotonicity holds when error is jointly optimized over weight decay and richness.  Please see the updated section 6.
>
> minor points:
>
> Both CIFAR10/MNIST and a CIFAR10/MNIST-derived binary classification task are included in the paper. What is the papers convention for differentiating between the binarized version and the original version? In places the paper appears to call the binary classification task “a CIFAR classification task” or “Binarized CIFAR10” but in others it then refers to this binary task just as CIFAR (I think this happens for example in line 370). Could the binary tasks have a name (for which CIFAR10 is a prefix perhaps) and then be referred to by that name to improve readability?
>
> > Thank you for catching this discrepancy.  To clarify, all random-feature ridge regression experiments are performed on the binarized CIFAR10 and MNIST tasks.  The deep ensemble experiment is on the standard CIFAR10 experiment.  We do not do MNIST classification with the original labels in any of our experiments.  We have updated the manuscript so that the binarized CIFAR10 and MNIST tasks are always referred to as such.
>
> The same notation of learning rate and task eigen structure, and for richness and the data-size-scaled degree of freedom parameters. Giving separate notation would be preferable.
>
> 118 – “(citations)” should instead be the actual citation
>
> 228 – Figure 1: Are the non-red lines supposed to be dashed in this plot? otherwise does the legend contain a dashed line? Since none appear in the actual plot.
>
> 340 – Figure 3 – could you draw the line for l* (= 1/(1+ alpha*(2*r – 1))) in red on the plots for figure 3 A?
>
> 397 – “a” is singular, but “tasks” is plural, should remove “a”?
>
> 452 – “that at the” -> “that the”
>
> > **Response**: Thank you for pointing out these typos and the error in the legend of figre 1.  We gave fixed them in our updated manuscript.  We appreciate your careful reading!  We will plan to add the lines for $\ell^*$ to the final version of figure 3A.
>
> Questions:
> In Figure 2c, ensembles (K>1) appear to be relatively robust to ridge parameter, where as K=1 is highly sensitive to it. Could this correspond to it potentially being more practically expedient to train small ensembles when optimally tuning the regularising parameter is expensive?
>
> > **Response**: This is an excellent point!  We have added a comment on the potential robustness of ensembleing methods in situations where fine-tuning hyperparameters is not feasible to our discussion of figure 3 in section 4.

---

### Author Response · Authors · 2024-11-21
**Updates to Rebuttal Version**

We thank all the reviewers for their time and valuable feedback.  Since our initial submission, we have made a number of updates to our manuscript which we will summarize here.  We will also respond individually to each reviewer's comments.

- We have updated our discussion of the RFRR risk formula to include a reference to [1], which provides a rigorous backing for the risk estimate we use.  We have also updated our terminology to refer to "Random Feature Ridge Regression (RFRR)" uniformly throughout the paper.
- We have added supplemental figures S3 and S6 confirming that the monotonic decreases in error described in theorems 1 and 2 for the MSE loss also hold for the binarized MNIST and CIFAR10 RFRR classification tasks at the level of the 0-1 test loss under majority vote and score-averaging.
- We have updated the methods we use to determine the "source" and "capacity" exponents for the binarized MNIST and CIFAR10 NNGP kernel regression tasks to the more robust methods described in [2], leading to a better fit between theory and experiment in fig. 4B.
- We have updated section 3.3 to elaborate on potentially complicating effects of ensembling which make Theorem 1 an interesting result, distinct from the $K=1$ case proven in [2].
- We have added a corrolary to section 4 which combines theorems 1 and 2 to guarantee that an ensemble of $K$ RFRR models with $N$ features each can only outperform a single RFRR model with $M$ features at optimal ridge if $NK>M$.  We confirm this for tasks with power-law structure and the binarized CIFAR10 RFRR task in a new supplemental figure S2.  This bound is tight in the overparameterized regime where $N \gg P$.
- We have updated the text of section 6 ("No Free Lunch from Feature-Learning Ensembles") to emphasize the importance of using $\mu P$ parameterization, to clarify the relationship between our theory and deep ensembles, and to simplify the discussion of our results.  In particular, we propose three conditions under which the "no free lunch from ensembles" principle holds in networks trained with $\mu P$ parameterization:
  - In the lazy training regime ($\gamma \to 0$) when the weight decay is tuned to its optimal value.
  - When weight decay and richness $\gamma$ are jointly tuned to their optimal values.
  - When richness $\gamma$ is fixed and training is performed *online* (i.e. without repeating data).
- We have updated Figure 5.A to sweep over a larger range of richness values $\gamma$, confirming that accuracy is monotonically decreasing with $K$ when weight decay and richness are jointly tuned to their optimal values.
- We have added experiments analogous to figure 5.A for CIFAR-10 classification but using the ResNet18 architecture.  We find that for this larger architecture, the "no free lunch from ensembles" principle still holds.


[1] Leonardo Defilippis, Bruno Loureiro, and Theodor Misiakiewicz. Dimension-free deterministic equivalents and scaling laws for random feature regression, 2024. URL https://arxiv. org/abs/2405.15699.

[2] James B. Simon, Dhruva Karkada, Nikhil Ghosh, and Mikhail Belkin. More is better in modern machine learning: when infinite overparameterization is optimal and overfitting is obligatory. CoRR, abs/2311.14646, 2023. doi: 10.48550/ARXIV.2311.14646. URL https://doi.org/ 10.48550/arXiv.2311.14646.

---

### Meta-Review · Area_Chair_x1Lb · 2024-12-20

**Metareview:**

The submission investigates whether ensembles can outperform single models when constrained to have the same total parameter count. This is undertaken from a theoretical point of view in the context of random feature ridge regression, and in the empirical context via neural networks. The results in both setting suggest that, when regularisation hyperparameters are optimal, the ensemble cannot outperform the single model. The reviewers generally agreed that the paper is easy to understand, even though the subject matter is quite complex. There were no concerns about the correctness of the theoretical contributions or the experimental setup. The main downsides that impacted my decision are the lack of theoretical novelty over several existing works, as pointed out by reviewer SSHY, and the mismatch between the experiments and theoretical analysis, as pointed out reviewer 6nYe.

**Additional Comments On Reviewer Discussion:**

There was quite substantial discussion between the authors and the two reviewers who gave negative scores. This primarily centred around the originality of the results and connection of the theoretical analysis to the empirical investigation. Reviewer SSHY argued fairly convincingly that the theoretical results do not provide much additional insight over the results given in Simon et al. (2023) and Bordelon et al. (2024) papers cited in the submission. This, coupled with the observation of 6nYe that the theory does not align so well with the experiments, are the main factors in my decision.

---

### Decision · Program_Chairs · 2025-01-22

Reject